

# Interference dynamics of matter-waves of SU($N$) fermions

Wayne Jordan Chetcuti[1,2,3], Andreas Osterloh[2], Luigi Amico[2,3,4,5∘] and Juan Polo[2]

**1** Dipartimento di Fisica e Astronomia, Via S. Sofia 64, 95127 Catania, Italy
**2** Quantum Research Center, Technology Innovation Institute, Abu Dhabi, P.O. Box 9639, UAE
**3** INFN-Sezione di Catania, Via S. Sofia 64, 95127 Catania, Italy
**4** Centre for Quantum Technologies, National University of Singapore,
3 Science Drive 2, Singapore 117543, Singapore
**5** LANEF 'Chaire d'excellence', Université Grenoble-Alpes & CNRS, F-38000 Grenoble, France

## Abstract

Interacting *N*-component fermions spatially confined in ring-shaped potentials display specific coherence properties. The orbital angular momentum per particle of such systems can be quantized to fractional values specifically depending on the particle-particle interaction. Here we demonstrate how to monitor the state of the system through homodyne (momentum distribution) and self-heterodyne system's expansion. For homodyne protocols, the momentum distribution is affected by the particle statistics in two distinctive ways. The first effect is a purely statistical one: at zero interactions, the characteristic hole in the momentum distribution around the momentum k=0 opens up once half of the SU(*N*) Fermi sphere is displaced. The second effect originates from the interaction: The fractionalization in the interacting system manifests itself by an additional 'delay' in the flux for the occurrence of the hole, that now becomes a characteristic minimum at k=0. We demonstrate that the angular momentum fractional quantization is reflected in the self-heterodyne interference as specific dislocations in interferograms. Our analysis demonstrates how the study of the interference fringes grants us access to both number of particles and number of components of SU(*N*) fermions.



## Contents

---

∘ On leave from the Dipartimento di Fisica e Astronomia "Ettore Majorana", University of Catania.

# 1 Introduction

Placed in the vacuum and spatially confined with suitable electromagnetic fields, ultracold atoms [1] feature robust coherence properties in the absence of cryostats. They can be realized in different physical conditions, like with a tunable atom-atom interaction or with fundamentally different quantum statistics of the gas constituents. Due to the remarkable progress in micro-optics technology, they can be trapped in a wide variety of potentials, shapes and intensities [2].

These are some relevant features as to why ultracold atoms provide an important instance of artificial quantum matter that can be used as 'hardware' both for quantum simulations and to advance the fabrication of quantum devices [3–5]. Atomtronics is the quantum technology of guided ultracold atoms: while the defining goal of the field is to fabricate quantum devices and sensors with enhanced performances, atomtronic circuits can define current-based quantum simulators probing quantum correlations in many-body systems [6, 7]. A natural venue for this research activity has been constructing analogs of electronic devices [8–11]. Atomtronics, though, has the potential to realize devices and simulators with new capabilities, relying on different physical properties compared with electronics. In the last decade, an intense activity has been devoted to bosonic matter-waves guided in circuits of a wide variety of shapes [6, 12–14]. Angular momentum quantization in $^{87}$Rb atomtronic ring-shaped circuits has been studied both theoretically and experimentally [15, 16]. Such studies have been instrumental in defining the atomic counterpart of SQUIDs [15–18], that are believed to be of paramount importance for guided interferometers [12, 19–22]. Recently, it has been predicted that attracting bosons can lead to an enhanced performance in rotation sensing [23, 24].

Recent advances in cold atoms experiments have re-kindled the interest in SU(*N*) fermionic systems [25–29]. These strongly interacting *N*-component systems, as provided by alkaline earth and ytterbium atoms, have an enlarged symmetry compared to SU(2) fermionic systems resulting in unique and and exotic physics [30, 31]. SU(*N*) fermions play a vital role in

a wide variety of contexts ranging from high precision measurement [32, 33] and quantum simulation [27, 28, 34] of many-body systems, to studying lattice confinement in high energy physics [35].

Here, we focus on the simple case of an atomtronic circuit provided by a ring-shaped quantum gas of SU($N$) fermions. In such circuits, a guided matter-wave, specifically a persistent current, can be generated by the application of an effective magnetic field [36–40]. Persistent currents in two-component ultracold fermions have been experimentally studied very recently [41, 42]. For $N$-component fermions confined in ring potentials, the theory predicts *a fractional quantization* of the orbital angular momentum per particle (henceforth referred to as angular momentum), with important differences arising on whether the atoms are subject to repulsive or attractive interaction [43, 44]. Such specific properties of quantization are expected to provide the core to fabricate quantum devices with enhanced sensitivity [23]. At the same time, these results re-affirm the notion that persistent currents can be used to define an instance of the aforementioned current-based quantum simulators for the diagnostic of interacting quantum many-particle system [6, 7, 45–47].

In this paper, we investigate the fractionalization of the persistent current flowing in an SU($N$) fermionic circuit through interference dynamics. Both homodyne and heterodyne protocols have been carried out so far. In homodyne protocols, the system of interest interferes with itself. Such logic has been widely employed in ultracold atoms experiments through time-of-flight (TOF) images of the atoms density for both bosons and fermions [6, 7, 17, 48–50]. Through this measurement technique, the *angular momentum quantization of the circulating current state can be monitored* [39, 51]. With heterodyne phase detection protocols, the phase portrait of the system flowing along the ring is obtained through its additional interference with a non-rotating quantum degenerate system placed at the center of the ring. This type of protocol has been experimentally realized both for bosons [52–54] and very recently for fermions [42]. The fringe pattern that arises is a spiral interferogram whose topological features (number of arms and dislocations) reflect the properties of a circulating current state [22, 42, 54, 55]. We employ both the homodyne and heterodyne protocols to analyze the interference dynamics of matter-waves of SU($N$) fermions. We demonstrate how the resulting interference patterns reflect important features of the system, including the specific angular momentum fractionalization and parity effects characterizing the system. Particularly, we highlight how our approach may be utilized to detect the number of particles $N_p$ and components $N$, both of which are notoriously hard to extract from an experimental setting [56].

The article is structured as follows. In Sec. 2 we introduce the physical system and the model. In Sec. 3 and Sec. 4, we present the results achieved for the momentum distribution and interferograms respectively. Conclusions and outlooks are presented in closing Sec. 5.

## 2 Methods

Consider $N_p$ SU($N$)-symmetric fermions, in a ring-shaped lattice composed of $L$ sites, pierced by an artificial magnetic flux $\phi$. The relevant physics of the model is captured by the SU($N$) Hubbard model [30, 31], which reads

$$\mathcal{H}_{\text{SU}(N)} = -t \sum_{j}^{L} \sum_{\alpha}^{N} \left( e^{i\frac{2\pi\phi}{L}} c_{j,\alpha}^{\dagger} c_{j+1,\alpha} + \text{h.c.} \right) + U \sum_{j}^{L} n_j n_j, \tag{1}$$

where $c_{j,\alpha}^{\dagger}$ creates a fermion with colour $\alpha$ on site $j$, and $n_j = \sum_{\alpha} c_{j,\alpha}^{\dagger} c_{j,\alpha}$ is the local particle number operator. The hopping amplitude and on-site interaction are denoted by $t$ and $U$ respectively. The presence of the flux is accounted for through the Peierls substitution $t \to t e^{i\frac{2\pi\phi}{L}}$.

The physics of the Hubbard model arises from the competition between the kinetic (hopping) and potential (interacting) terms. As such, the ratio between the hopping and interaction parameters dictates the physics that we observe in our system. For strong attractive interactions ($|U| \gg t$), SU($N$) fermions are able to form bound states of different types and nature, which in turn causes part of the particles to localize together, while still adhering to the Pauli exclusion principle [44, 57–59]. On the other hand, strong repulsive interactions ($U \gg t$) causes the fermions to be restricted in place in different lattice sites. This is evidently visible for fillings $N_p/L = 1$, where in the case of SU($N > 2$) fermions a phase transition occurs at a critical value of $U$ from a superfluid to a Mott phase [31, 43]. Recently, the physics of the SU($N$) Hubbard model was explored for both attractive [44] and repulsive [43] interactions by utilizing the persistent current, the response of the system to the applied field in model (1), as a diagnostic tool. The persistent current is a matter-wave current defined as $I = -\partial E_0 / \partial \phi$ with $E_0$ being the ground-state energy. According to Leggett's theorem, the ground-state energy and consequently its derivative, the persistent current, display periodic oscillations in the flux, with a period fixed by the elementary flux quantum $\phi_0 = \hbar/mR^2$ with $m$ and $R$ denoting the atoms' mass and ring radius respectively [60]. Therefore, a change in the period $\phi_0$ gives crucial information about the physical nature of the system, in the same spirit as current-voltage characteristics in solid state physics (see [23, 43–46, 61] for the persistent current as a probe for bosonic and fermionic systems).

In cold atoms platforms, several features of the persistent current can be readily observed through time-of-flight imaging of the density distributions of the gas at large times, after it is released from its trap confinement and allowed to expand [6, 7, 17, 39, 62]. Such an image corresponds to the momentum distribution of the system at the moment in which it is released from the trap. For the different SU($N$) species, the latter quantity reads

$$n_\alpha(\mathbf{k}) = |w(\mathbf{k})|^2 \sum_{j,l} e^{\iota \mathbf{k}(\mathbf{r}_l - \mathbf{r}_j)} \langle c_{l,\alpha}^\dagger c_{j,\alpha} \rangle, \tag{2}$$

with $w(\mathbf{k})$ being the Fourier transform of the Wannier functions and $\mathbf{r}_j$ denotes the position of the lattice sites in the ring's plane (see Sec. A.1 for the derivation). Also important for our analysis is the variance of the momentum distribution $\sigma_{n_k}^{(\alpha)}$, given by $\sigma_{n_k}^{(\alpha)} = \sqrt{\langle n_\alpha^2 \rangle - \langle n_\alpha \rangle^2}$.

For bosons [22] and quite recently for fermions [55], it has been shown that interference effects can be observed by considering higher order density-density correlations. In this setup, one starts with an annular ring under study and a central degenerate system at rest, both of which are uncoupled initially. Due to this, as both ring and center are released from their confinement and start to interfere, eventually there is an uncertainty when measuring two or more particles, about whether these particles originated from the center or the ring. As the uncertainty about the particles' origin increases, one gains more certainty about the phase [22, 63]. Therefore, inline with the previous protocols for bosons and fermions, we focus on the density-density correlator

$$G(\mathbf{r}, \mathbf{r}', t) = \sum_{\alpha, \beta}^{N} \langle n_\alpha(\mathbf{r}, t) n_\beta(\mathbf{r}', t) \rangle, \tag{3}$$

to observe interference patterns in SU($N$) fermions. Seeing as the interference that arises is provided by the cross-terms involving contributions from both the center and the ring [55], we calculate and focus on the center-ring correlations whose expression reads

$$G_{R,C} = \sum_\alpha \sum_{j,l} I_{jl}(\mathbf{r}, \mathbf{r}', t) \langle c_{l,\alpha}^\dagger c_{j,\alpha} \rangle, \tag{4}$$

where $I_{jl}(\mathbf{r}, \mathbf{r}', t) = -w_c(\mathbf{r}', t) w_c^*(\mathbf{r}, t) w_l^*(\mathbf{r}' - \mathbf{r}_l', t) w_j(\mathbf{r} - \mathbf{r}_j, t)$.

Our approach utilizes exact diagonalization and DMRG [64, 65], whenever possible, to evaluate and analyze the correlators. Here, the energy scale is given by fixing $t = 1$ and $\phi_0 = 1$. In our analysis, we only consider systems with an equal number of particles per colour.

# 3 Momentum distribution of SU($N$) fermions

The momentum distribution of particles on a ring is one of the few observables of the momenta that can be experimentally probed [17, 37, 39]. This homodyne protocol is of particular interest in the field of atomtronics, since the persistent current is visible in experiments by studying the particles' momentum distribution [6, 7, 51]. In the case of coherent neutral matter circulating in a ring with a given angular momentum quantization, a characteristic hole is observed in the momentum distribution [51, 66]. On the other hand, no hole is observed at the same flux when there is a reduced coherence, e.g. for attractive interactions [23, 44, 55, 61]. Nonetheless, looking at the variance of the momentum distribution, one is still able to observe the corresponding angular momentum quantization. Here, we make an in-depth analysis of the momentum distribution of SU($N$) fermions for both attractive and repulsive regimes. As we will see, the distinct physical features and characteristics of these two regimes can be aptly captured through homodyne interference images.

## 3.1 Free particles

The momentum distribution of spinless fermions at zero interactions is a sum of discrete Bessel functions $\sum_{\{n\}} J_n(\mathbf{k})$ of order $n$, with $\mathbf{k}$ denoting the momentum and $n$ being the quantum numbers of the levels the particles occupy [18, 55]. From this expression, it is clear that the momentum distribution is dependent on the sets of $n$, with the ground-state configuration being such that they are distributed symmetrically around zero (see Sec. A.2). These quantum numbers are related to the angular momentum per particle $\ell$ [67, 68].

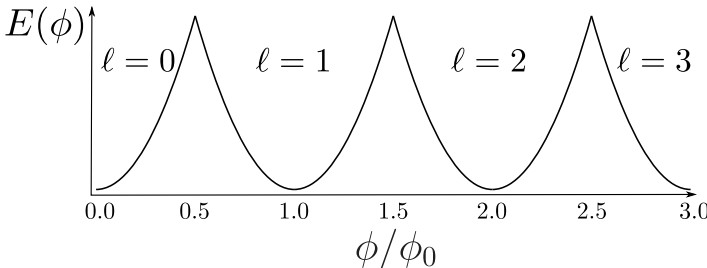

Figure 1: Energy as a function of the effective magnetic flux $\phi$, denoted by $E$ and $\phi$ respectively. As one crosses from one parabola to the next with increasing $\phi$, the angular momentum quantum number $\ell$ increases.

Apart from the zeroth order Bessel function, all other orders are zero-valued at $\mathbf{k} = 0$ –see Fig. 2. Consequently, when the particles inhabit the $n = 0$ level, corresponding to the zeroth order Bessel function, the momentum distribution is always peaked at the origin. Such is the case for $\ell = 0$ in Fig. 2. When threaded by an effective magnetic flux, the ground-state energy displays periodic oscillations characterized by a given angular momentum $\ell$. As the flux increases and we move from one energy parabola with a given $\ell$ to the next, the quantum numbers $n$ need to be changed to counteract the increase in flux and minimize the energy (see Sec. A.2). Eventually, the set of $\{n\}$ is such that no spinless particles inhabit the $n = 0$ level at a given value of $\ell$. Being a sum of discrete Bessel functions, the momentum distribution

becomes zero-valued at the origin and a hole opens up [55]. The value of the angular momentum $\ell$ needs to be such that Fermi sphere is displaced by the ceiling function $\lceil \frac{N_p}{2} \rceil$ [51,55] (see Sec. A.2 for a schematic figure). Therefore, there is a 'delay' in observing the hole with increasing $N_p$. This needs to be contrasted with bosons in a Bose-Einstein condensate, which due to the different statistics all reside in the $n = 0$ level at $\ell = 0$. In turn, there is no 'delay' for the characteristic hole, which opens up at $\ell = 1$ irregardless of $N_p$.

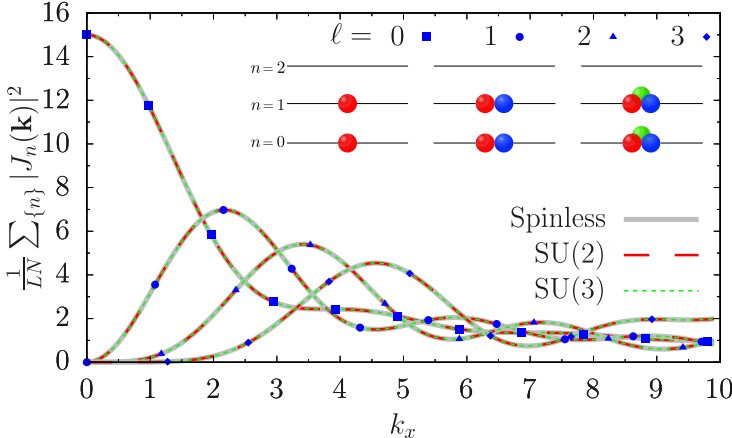

Figure 2: Main figure depicts the momentum distribution $\sum_{\{n\}} |J_n(\mathbf{k})|^2$ re-scaled by the number of components $N$ against $k_x$. For the $\ell = 0$ configuration, the momentum distribution has a finite value at the origin. However for $\ell > 0$, the function collapses to zero at the origin. Insets depict the ground-state configuration $\ell = 0$ for spinless, SU(2) and SU(3) fermions containing 2,4,6 particles respectively at $U = 0$.

The same logic used for spinless fermions applies when one considers SU($N$) fermions. On increasing $N$, the restriction imposed on the system by the Pauli exclusion principle relaxes: $N$ particles can occupy a given level –see Fig. 2. Accordingly, the Fermi sphere needs to be displaced less with increasing $N$. We find that for a system with $N_p$ SU($N$)-symmetric fermions, a hole in the momentum distribution opens up when we displace by $\lceil \frac{N_p}{2N} \rceil$. In other words, the angular momentum required for a hole to open up is $\ell_H = \lceil \frac{N_p}{2N} \rceil$ and $\phi_H$ is the flux at which one transitions to the energy parabola with this corresponding angular momentum. For $N_p < N$, all particles will reside in the $n = 0$ level. Indeed, as $N \to \infty$, SU($N$) fermions behave as bosons with regards to the level occupation.

Systems with an equal and commensurate value of $W = \frac{N_p}{N}$ display similar features. The persistent current's parity is one such feature whereby it is diamagnetic [paramagnetic] if $N_p = (2m + 1)N$ [$N_p = (2m)N$] with $m$ being an integer [43,60]. Likewise, we have that the momentum distribution scaled by $1/N$ is the same for equal and commensurate $W$ –Fig. 2. Clearly, this is not the case when $W$ is commensurate but not equal for different SU($N$), such as systems of $N_p = 6$ fermions with SU(2) and SU(3) symmetry for example. Owing to the different particle occupations, the momentum distribution and consequently the shifts in the sets of $\{n\}$ and angular momentum $\ell_H$ for a hole to appear is different.

## 3.2 Interacting particles

Having established the basis for the momentum distribution for SU($N$) fermions at zero interactions, we now turn our attention to systems with repulsive and attractive interactions. At small values of the interaction, one observes the same features as in the free fermion cases described above (see also [55] for SU(2)). Here, we focus on the regimes of intermediate and infinite interactions.

### 3.2.1 Repulsive interactions

In the case of fermionic systems with repulsive interactions, the persistent current displays fractionalization with a reduced period dependent on the number of particles $N_p$ irrespective of the number of components $N$ [43, 69]. The fractionalization originates from energy level crossings between the ground and excited states that create spin excitations to counterbalance the increase in the effective magnetic flux. In other words, we go from the initial parabola observed at $U = 0$ with a period $\phi_0$, to $N_p$ piece-wise parabolas/peaks with a reduced period of $\phi_0/N_p$ (see Sec. A.2). This effect results in a momentum distribution depression at infinite repulsion. In contrast with the characteristic hole, the depression is not zero-valued at the origin but is a local minimum, i.e. a dip in the momentum distribution –Fig. 3 (**a**). Apart from cases like the one depicted in Fig. 3 (**a**), we generally find a non-monotonous behaviour in the peak of the momentum distribution (see Sec. A.2).

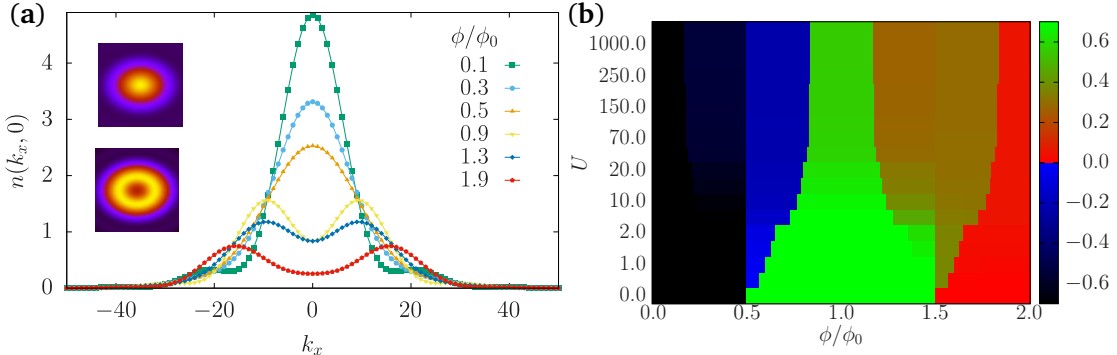

Figure 3: (**a**) Cross-section of the momentum distribution $n(k_x, 0)$ at strong repulsive interactions $U = 10,000$ for various values of the effective magnetic flux $\phi$. When the threshold imposed by the fractionalization is surpassed, the momentum distribution collapses at $k_x = 0$ and a depression is observed. For larger values of $\phi$, the depth of the depression increases. (**b**) Second derivative of the momentum distribution $n(k_x, k_y)$ evaluated at $k_x = k_y = 0$, defined as $\frac{\partial^2 n(k_x, 0)}{\partial k_x^2}\big|_{k_x=0}$, as a function of $\phi$ and $U$. A change in the sign of the second derivative denotes going from a peak (negative) to a depression (positive) that corresponds to a change in colour from blue to green respectively. At $U = 0$ the hole opens up at $\phi = 0.5$. On increasing the interaction, the depression appears at larger $\phi$ thereby reflecting the fractionalization in the system. As $U \rightarrow \infty$, the system achieves complete fractionalization and the flux at which a depression is displayed corresponds to $\phi_D$. Note that the $y$-axis is not linear in values of the interaction. Results were obtained with exact diagonalization for $N_p = 3$ SU(3) symmetric in $L = 15$ sites.

Interestingly enough, when the persistent current is fractionalized, the depression appears at fluxes $\phi_D$ that are found to be significantly larger than the flux values corresponding to the cases in which the angular momentum is quantized to integer values. In other words, the fractionalization in the system causes a specific 'delay' in observing the momentum distribution depression –see Fig. 3. We remark that this 'delay' is an add-on to the one observed at zero interactions. It is solely dependent on $N_p$ and independent of $N$ due to the nature of the fractionalization. The depression in the momentum distribution occurs at $\phi_D = \phi_H + \frac{N_p-1}{2N_p}$ where $\phi_H$ is the flux at which a hole appears at zero interaction (see Sec. A.2). For intermediate interactions, where the persistent current is partially fractionalized, we track the depression by calculating the second derivative of the momentum distribution as a function of $\phi$ and $U$– Fig. 3 (**b**).

### 3.2.2 Attractive interactions

Just like in its repulsive counterpart, the persistent current also fractionalizes for attractive interactions. For strong attractive interactions, the formation of bound states is reflected by a reduced period of the current $\phi_0/r$ with $r$ being the number of particles in the bound state [44]. Even though the state does acquire current, the formation of bound states with reduced coherence, drastically reduces the visibility of the hole in the momentum distribution –Fig. 4 (a).

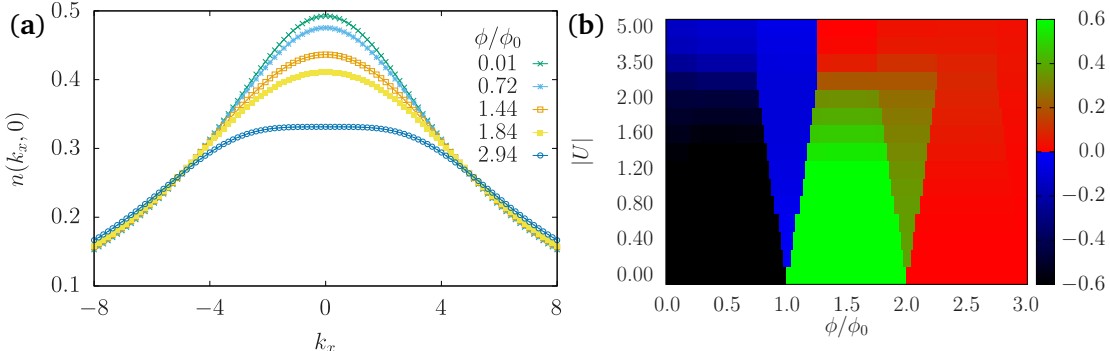

Figure 4: (**a**) Cross-section of the momentum distribution $n(k_x, 0)$ for $N_p = 6$ fermions with SU(3) symmetry for various values of the effective magnetic flux $\phi$ at $U = -5$. The momentum distribution is always peaked for any value of $\phi$ due to the reduced coherence in the system. As $\phi$ increases, the height of the peak decreases in a monotonous behaviour. (**b**) Second derivative of the momentum distribution $n(k_x, k_y)$ evaluated at $k_x = k_y = 0$, defined as $\frac{\partial^2 n(k_x, 0)}{\partial k_x^2}\big|_{k_x=0}$, as a function of $\phi$ and $U$ for $N_p = 4$ SU(2) symmetric fermions. A change in the sign of the second derivative denotes going from a peak (negative) to a depression (positive) that corresponds to a change in colour from blue to green respectively. In addition to the delay observed for strong interactions as in the repulsive case, one observes that the depression is smoothed out with increasing interactions. Results were obtained with exact diagonalization for $L = 15$.

Nevertheless, the variance of the width of the momentum distribution $\sigma_{nk}$ has been demonstrated to be a figure of merit for fractional currents [23, 44] –Fig. 5. We find that the variance is not a good measure when it comes to the repulsive case since the peak does not display monotonous behaviour (see Sec. A.2). The partial fractionalization of the persistent current at intermediate interactions can also be tracked through the second derivative of the momentum distribution – Fig. 4 (**b**). It should be noted that upon formation of the bound state, the depression is washed out.

## 4 Self-heterodyne interferograms of SU($N$) fermions

Properties of circulating current states can be detected through self-heterodyne phase detection protocols [53, 54, 70]. In these protocols, the ring is made to interfere with a quantum degenerate system at its center. During TOF expansion, the combined wavefunction (ring and center) evolves in time and interferes with itself giving rise to characteristic spiral interferograms [22, 53, 55]. Topological features of the spiral pattern reveal information on the current's orientation and intensity. Furthermore, the angular momentum quantization of the current (quanta of rotations) is given by the number (or the order) of spirals and their orientation.

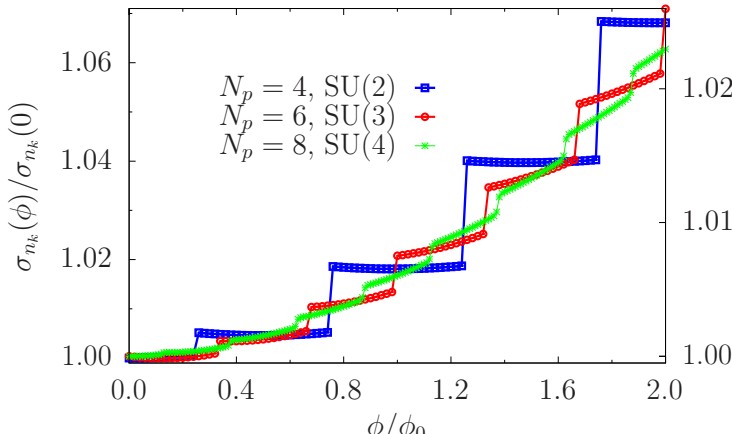

Figure 5: Variance of the momentum distribution, $\sigma_{n_k}(\phi)$, against the effective magnetic flux $\phi$ for $N_p = 4$ (blue), $N_p = 6$ (red) and $N_p = 8$ (green) fermions with SU(2), SU(3) and SU(4) symmetry respectively for attractive interactions. On going to infinite attraction the persistent current fractionalization depends on the number of components, which is reflected in the variance of the width of the momentum distribution. For SU(2) fermions, the variance has two steps in one flux quantum ($\phi/\phi_0 = 1$). Going to SU(3) (SU(4)) fermions, we have that the period is reduced by 1/3 (1/4) upon fractionalization, which is reflected by the variance having 3 (4) steps. Results for SU(2) and SU(3) cases were obtained with exact diagonalization, with DMRG being employed for the SU(4) case. The parameters used were $U = -5$ and $L = 15$, with $U = -3$ for the SU(4) case.

Just like in the momentum distribution, the particles' statistics are reflected in the interference patterns. Due to the Pauli exclusion principle, fermionic particles occupy distinct levels having different momenta associated to them. Hence, when the fermions start to circulate, the imparted phase gradient of the wavefunction couples to the various momenta. These different phases recombine giving rise to the spiral-like interference. The multiple momenta contributing to the interference pattern as well as the particle's statistics results in dislocations (radially segmenting lines) [55] (see Fig. 6). Previously, we highlighted how the momentum distribution at zero interaction displays similarities for systems with equal $W = N_p/N$. Seeing as the features of the interferogram depend on the particle distribution, then one would expect that systems with commensurate and equal $W$, display interferograms with the same characteristics. In the following, we build up on the recent work on interferograms of SU(2) fermions at zero and weak interactions [55], by generalizing to SU($N$) fermions and extending the analysis to the intermediate and strongly interacting regimes. We remark that DMRG cannot be applied to large interactions. Indeed, it is widely known that it has issues with convergence in this regime. Furthermore, the problem is exacerbated by multi-degenerate ground-states of SU($N > 2$) [43]. Consequently, we opted to carry out the self-heterodyne analysis by considering SU(2) systems that can be addressed by exact diagonalization. Nonetheless, this does not hinder our analysis since as we shall see, interferograms with equal $W$ display similar characteristics.

## 4.1 Free particles

At zero interaction and short time expansion one observes spiral-like patterns, along with dislocations in these interferograms [55]. These dislocations arise from the various momenta components contributing to the expansion and indicate the suppression of the particle density

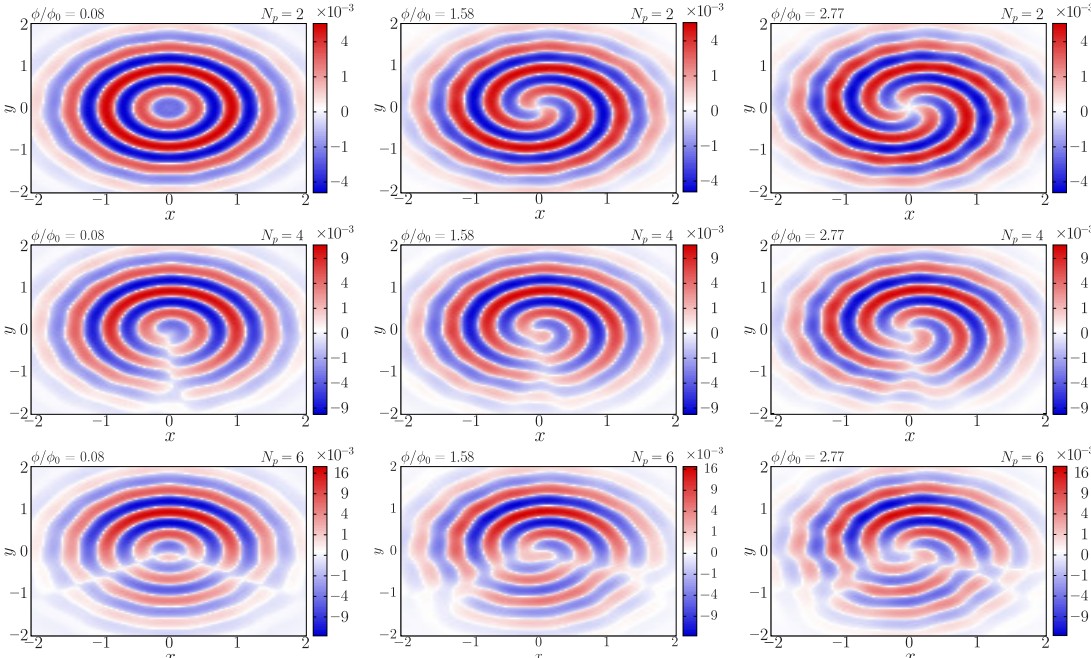

Figure 6: The interference $G_{\mathrm{R,C}}$ between ring and center for $N_p = 2, 4, 6$ SU(2) symmetric fermions, is shown as a function of the effective magnetic flux $\phi$ at zero interaction and short time $t = 0.033$. For $N_p = 2$ (top) and $N_p = 4$ (middle), the number of spirals is an indication of the angular momentum quantization $\ell$. Note that at $\phi = 1.58$, the difference in spirals between the two cases arises due to the different parity of the two systems, that correspond to a degeneracy point of $\phi = s + 1/2$ and $\phi = s$ respectively, with $s$ being an integer. In contrast to the $N_p = 2$ case, we observe dislocations (segmenting lines) in the interference patterns for $N_p = 4, 6$. Lastly, by comparing $N_p = 2$ and $N_p = 6$ at $\phi = 1.58$ we observe that the latter displays only one spiral instead of two like its counterpart. The reason being that at $U = 0$ for $N_p = 6$, the hole opens up at $\phi = 1.5$. All correlators are evaluated with exact diagonalization for $L = 15$ by setting $\mathbf{r}' = (0, R)$ and radius $R = 1$. The color bar is non-linear by setting $\mathrm{sgn}(G_{\mathrm{R,C}})\sqrt{|G_{\mathrm{R,C}}|}$.

at these positions in space. The number of dislocations corresponds to $W-1$ –see Figs. 6 and 7. These dislocations do not depend on the flux threading the system.

For bosons, the number of spirals gives an indication on the number of rotations, or rather the angular momentum quantization $\ell$ of the current [22, 54]. However, it is not as straightforward when it comes to fermions. Owing to their different statistics, the level occupation of fermions is broader than that of bosons. In order for spirals to emerge in the interference patterns, the given system of fermions needs to displace its Fermi sphere by $\lceil \frac{N_p}{2N} \rceil$, which corresponds to the characteristic hole in the momentum distribution. After this, the number of spirals grows with increasing angular momentum.

When $W = 1, 2$, the number of spirals reflect the angular momentum quantization (since these cases only require one Fermi sphere 'shift' like in bosons) –see upper and middle panels of Fig. 6. For $W > 2$, the number of observed spirals is not indicative of the angular momentum quantization –see lower panel in Fig. 6. By keeping the number of particles fixed and increasing the number of components, enables more particles to inhabit a given level. Finally for $N > N_p$, SU($N$) fermions behave as bosons with regards to level occupations –Fig. 7. Additionally, we have that for equal and commensurate $W$, interferograms display similar features –see Figs. 6 and 7.

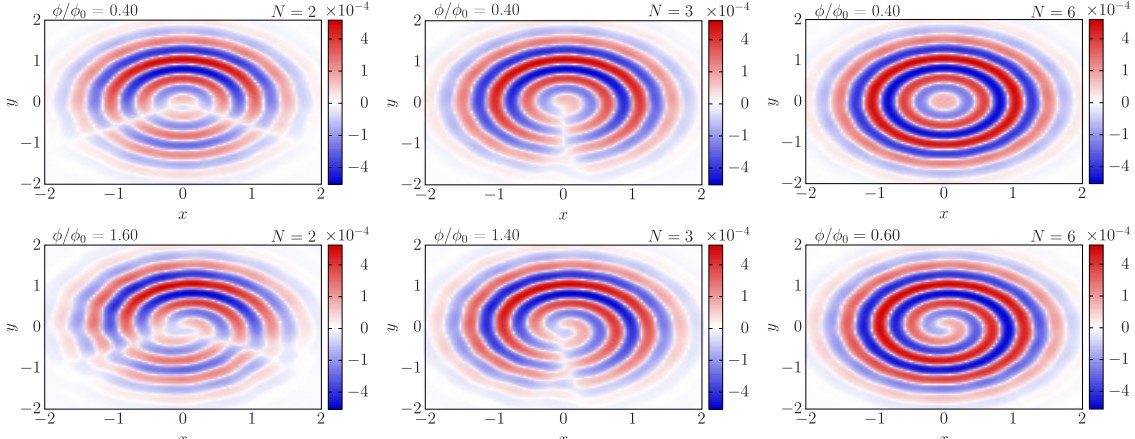

Figure 7: The interference $G_{R,C}$ between ring and center for $N_p = 6$ particles with $N = 2, 3, 6$, as a function of the effective magnetic flux $\phi$ at $U = 1$ and short time $t = 0.025$. In the top panel, the number of dislocations in the interferograms is 2,1,0 for $N = 2$ (left), $N = 3$ (middle) and $N = 6$ (right) respectively. The bottom panel depicts the value of $\phi$ where a spiral is observed. On increasing $N$, the spiral appears at a lower value of $\phi$ since the displacement of the Fermi sphere is inversely proportional to $N$. All correlators are evaluated with DMRG for $L = 15$ by setting $\mathbf{r}' = (0, R)$ and radius $R = 1$. The color bar is non-linear by setting $\text{sgn}(G_{R,C})\sqrt{|G_{R,C}|}$.

## 4.2 Interacting particles

### 4.2.1 Repulsive interactions

For a system with infinite repulsive interactions, one is able to not only track but also observe the fractionalization through the self-heterodyne phase portrait. Remarkably, the fractional angular momentum, which corresponds to different spin excitations [43,69], is explicitly captured in the interferogram through the dislocations that are now dependent on the flux. Indeed, the dislocations are able to characterize the $N_p$ fractionalized parabolas due to the different types of spin excitations through the number and orientations of the dislocations – Fig. 8. Specifically, there are $\lceil \frac{N_p}{2} \rceil + 1$ different types of interference patterns (i.e. different number and orientation of the dislocations). The characteristic spirals are also observed in interferograms at infinite repulsion. However, we remark that due to the dislocations, it is very hard to deduce the order of the spirals. We find that in order for a persisting spiral to emerge, the magnetic flux $\phi$ needs to exceed $\phi_{S_+} = \phi_H + \frac{N_p-1}{2N_p}$ (see Sec. B.1). This is in line with the appearance of the depression in the momentum distribution.

In the intermediate case, the interference patterns capture the partial fractionalization in the system. Indeed, on going from zero to infinite interactions as in Fig. 9, we observe the change in the orientation and number of dislocations as the system undergoes fractionalization.

### 4.2.2 Attractive interactions

In contrast to the homodyne protocol, the self-heterodyne one provides direct information on the fractionalization of the persistent current. Firstly, the $N$ parabolas that originate due to the fractionalization are characterized by different dislocations, both in number and orientation. Additionally, spirals emerge in the interferogram. Interestingly enough for $N$-body bound states, the observation of the spiral occurs at $\phi_{S_-} = \phi_H + \frac{N-1}{2N}$, which would correspond to when one would expect the momentum distribution depression (see Sec. B.2). Just like

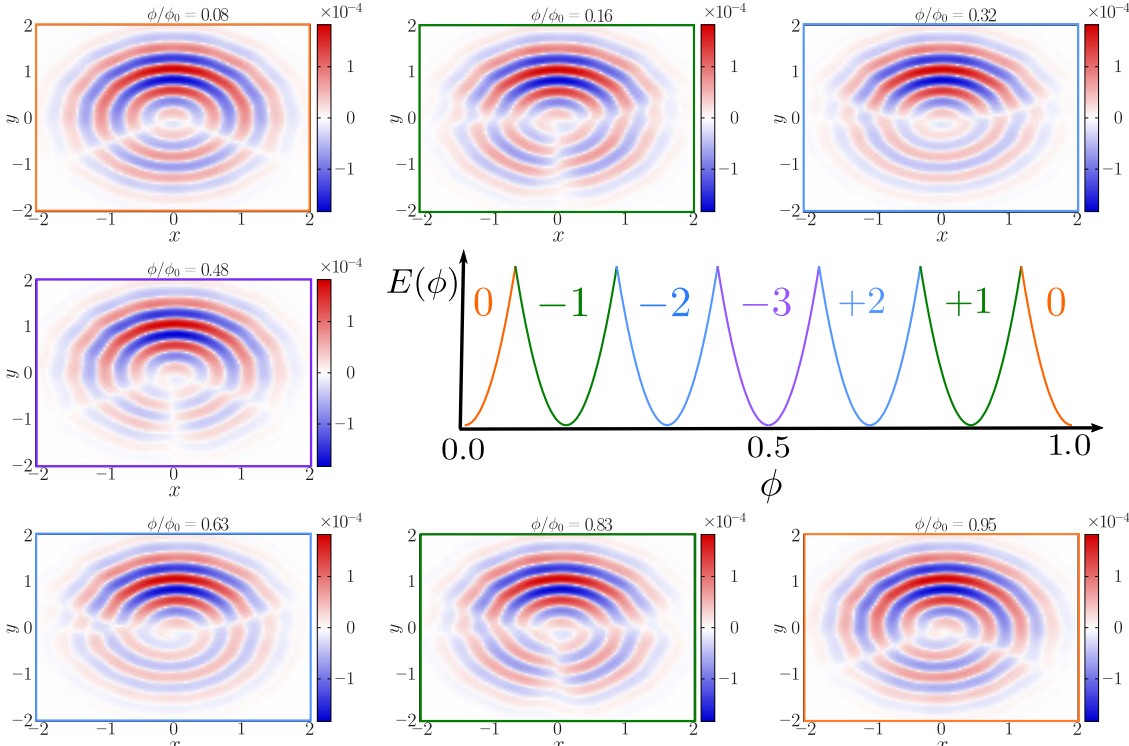

Figure 8: The interference $G_{R,C}$ between ring and center for $N_p = 6$ particles with SU(2) symmetry against the effective magnetic flux $\phi$ at $U = 1000$ and short time $t = 0.025$. Middle right panel is the schematic for the fractionalized energy parabolas at infinite repulsive interactions for the corresponding system (numbers on parabola correspond to the spin quantum numbers in the Bethe Ansatz solution [43]). The interference pattern corresponding to the first parabola (orange), displays two dislocations as in the zero interaction case. Moving to the next parabola (green), where we have the generation of a spin excitation, there are three dislocations. Traversing the other parabolas, we observe that apart from the number of dislocations, the orientation changes as well –see upper left and right panels where we observe an downward and upward V-shape respectively. By comparing each interference pattern to each energy parabola, we see that there is a symmetry around $\phi = 0.5$ that reflects the fact the the energy $E$ is symmetric around this point. Indeed, the number of different interference patterns corresponds to $\lceil \frac{N_p}{2} \rceil + 1$. Note that there is no spiral in the bottom right intereferogram since no hole has opened up. All correlators are evaluated with exact diagonalization for $L = 15$ by setting $\mathbf{r}' = (0, R)$ and radius $R = 1$. The color bar is non-linear by setting $\mathrm{sgn}(G_{R,C})\sqrt{|G_{R,C}|}$.

in the repulsive case, there is a 'delay' in visualizing the spirals. However, in this case the fractionalization is dependent only on $N$ and as such the 'delay' associated to it is uniform, irrespective $N_p$.

We remark that the mentioned results pertain only to SU(2) fermions. For SU($N > 2$), an interferogram is not adequate to capture any information about the attractive system. The self-homodyne interference patterns rely on the use of a two-body correlator, which is an accurate measure when one has bound pairs as in SU(2) fermionic systems. However, bound states consisting of a larger number of particles, such as trions in the SU(3) case, probably require an $N$-body correlator to adequately describe the system.

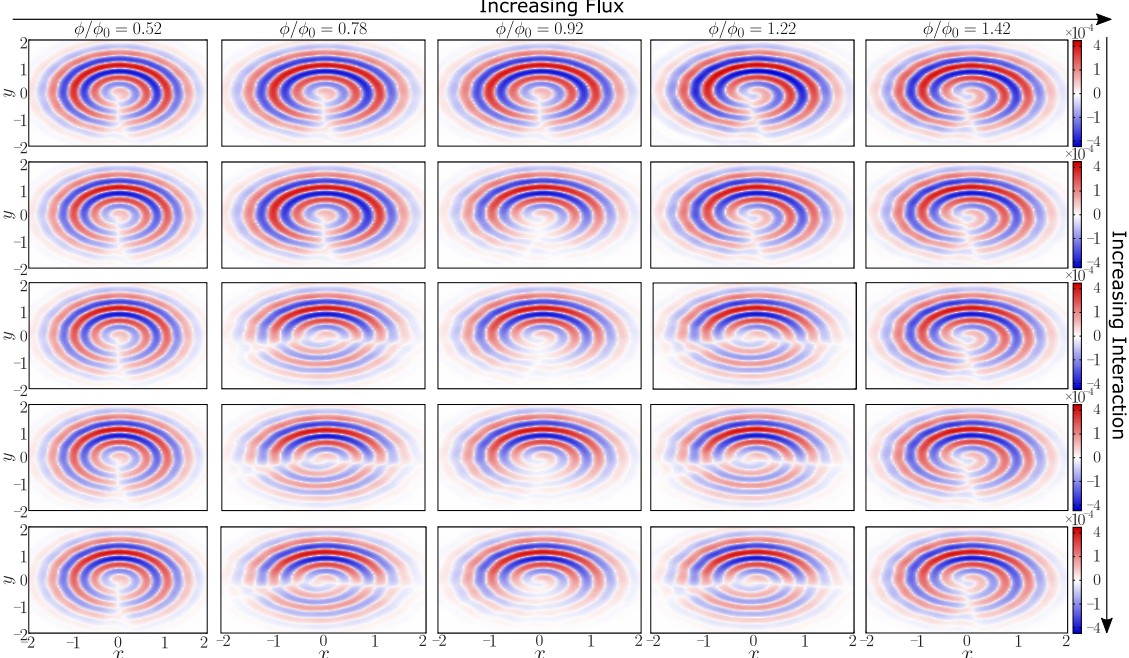

Figure 9: The interference $G_{R,C}$ between ring and center for $N_p = 4$ particles with SU(2) symmetry against the effective magnetic flux $\phi$ at short time $t = 0.026$ as a function of the interaction $U$. In this figure, one can clearly see how the number and orientation of the dislocations change as one increases the interaction. All correlators are evaluated with exact diagonalization for $L = 15$ and repulsive $U = \{0, 10, 20, 50, 5000\}$ (panels are in descending order) by setting $\mathbf{r}' = (0, R)$ and radius $R = 1$. The color bar is non-linear by setting $\mathrm{sgn}(G_{R,C})\sqrt{|G_{R,C}|}$.

## 5 Discussions and Conclusions

In this work, we utilize the persistent current to analyze and characterize different SU($N$) fermionic systems. To this end, we investigate the interference dynamics via both homodyne (momentum distribution) and heterodyne (co-expansion of two concentric condensates) protocols by applying a combination of exact diagonalization and whenever possible DMRG and Bethe Ansatz. Both of them can be experimentally probed within the cold atoms infrastructure and current know-how.

**Free particle regime –** For spinless fermions, the characteristic hole in the momentum distribution, reflecting coherent matter-wave flow, opens up when the effective magnetic flux displaces half of the Fermi sphere [55]. In the following, we will refer to such a feature as a 'delay' in the value of the flux at which a hole occurs. For $N$-component fermions, the Pauli exclusion principle relaxes and allows more particles to inhabit a given level –see Fig. 2. Accordingly, we find that a hole in the momentum distribution appears if the Fermi sphere is displaced less, precisely by the ceiling function $\lceil \frac{N_p}{2N} \rceil$. This is consistent with the fact that SU($N$) fermions (with $N_p < N$) resemble bosons for $N \to \infty$. We find that the features of the momentum distribution are consistent with the heterodyne interference images. In particular, holes in the momentum distribution correspond to spirals in the interferograms. Note that, in contrast with the bosonic case [22,52–54], we find that the order of the spirals does not reflect the angular momentum quantization of the system. Furthermore when the number of components $N$ divides the number of particles $N_p$, $\frac{N_p}{N} - 1$ dislocations (radially segmenting lines)

appear in the interferograms giving information about the number of particles present in the system –see Fig. 6. These observations still hold true in the case of small attractive or repulsive interactions [55]. Moreover, we note that at zero interactions the properties displayed by the homodyne and self-heterodyne phase portraits depend solely on $\frac{N_p}{N}$ –see Figs. 6 and 7.

**Repulsive regime –** The persistent current fractionalizes with the bare flux quantum $\phi_0$ reduced by $N_p$ (implying that the energy is periodic with a period of $\phi_0/N_p$). Fractionalization originates through level crossings between the ground and excited states that result in the creation of spin excitations in the ground-state [43]. Interestingly enough, we find that the fractional values of the angular momentum are not displayed as holes in the momentum distribution as in the non-interacting case. In addition, as soon as the correlations depart from the non-interacting case, the characteristic hole becomes a small depression (finite valued local minimum) in the momentum distribution at momentum $\mathbf{k} = 0$ –Fig. 3 (**a**). Moreover, an additional 'delay' characterising the specific fractionalization of the angular momentum is found. In other words, the depression occurs at a larger flux value than the one where the hole opens up –Fig. 3 (**b**). Specifically, for a given system with $N_p$ particles, a momentum distribution depression appears at $\phi_D = \phi_H + \frac{N_p-1}{2N_p}$, where $\phi_H$ is the flux at which the hole opens up at zero interactions and the second term accounting for the fractionalization 'delay'. Therefore, such a property makes $N_p$ accessible by monitoring the actual value of the flux at which the depression occurs. At intermediate interactions, the system experiences only partial fractionalization. Nonetheless, this is captured by the homodyne protocol, by tracking the momentum distribution depression at zero momenta.

Heterodyne interferograms embody the features of the fractionalization. Apart from observing the 'delay' through the emergence of the spiral as in the zero interaction case, the angular momentum fractionalization can be tracked by monitoring the number and orientation of the dislocations: The presence of the spin excitations modifies the dislocations that are observed at zero interaction –see Figs. 8 and 9. In contrast with the zero interaction case where the dislocations are not flux dependent, here the dislocations are dependent on the flux, enabling us to monitor the spin excitations in the system through interference patterns.

**Attractive regime –** Like its repulsive counterpart, a system with attractive interactions also experiences fractionalization, dependent on the number of components $N$. However, the fractionalization is not readily observed in the momentum distribution, at least not directly. Due to the reduced coherence that transpires from the formation of bound states, no depression is observed –Fig. 4. As such, one cannot monitor the 'delay' in the occurrence of the aforementioned depression. Nonetheless, by measuring the variance of the width of the momentum distribution one can deduce the SU($N$) nature of the system through the number of steps depicted, but no information regarding the number of particles can be obtained –Fig. 5. In the intermediate regime of interaction, we observe that the characteristic depression gets 'delayed' to larger values of the effective magnetic flux as the current gets fractionalized. Eventually, as the interaction strength increases, the depression is smoothed out.

For SU($N > 2$) systems self-heterodyne interference patterns calculated via density-density correlators are found to be incapable of providing observables to monitor the persistent current pattern. In fact, a higher order correlator may be required in order to capture the features of $N$-body bound states. In the case of SU(2) systems, we find that the fractionalization is characterized by a change in the number of dislocations in the interferograms. Remarkably, the flux values where a spiral emerges corresponds to the ones that would result in a depression (that is suppressed in this regime) in the momentum distribution. Just like in its repulsive counterpart, the spirals experience a two-fold 'delay' originating from the combined effect of displacement of the Fermi sphere and the fractionalization.

Self-heterodyne interference patterns for SU(2) attractive fermions were recently observed experimentally in the context of the BCS-BEC crossover [42]. Specifically, the interference fringes they observe correspond to BEC side of the crossover. Our analysis, instead, pertains to the BCS regime, where our results predict a characteristic 'delay' that provides information on the structure of the Fermi surface and the number of components involved in the bound state. The interference patterns within the BCS regime still remain to be analysed experimentally, with the main challenge lying in the fast expansion stemming from the large momenta occupations of the fermions. One route to address present limitations involves using dilute density systems such that the particles do not occupy levels with high momentum. Another option would be to selectively address particles close to the Fermi surface and perform the expansion.

Lastly, we point out that a generalized version of the BCS-BEC crossover for SU($N > 2$) fermions can be addressed through persistent currents. Essentially, one can treat an $N$-body bound state of attractive multicomponent fermions as composite fermions or bosons, described by spinless fermions with $p$-wave interactions or super Tonks Girardeau phase of attracting bosons for odd and even $N$ respectively [71]. The parity of the persistent current flowing in such systems (retains) loses its parity, i.e. be it diamagnetic or paramagnetic, if the wavefunction is (anti-)symmetric [44]. In the case of SU(2) fermions such an analysis was carried out in [61] with self-heterodyne interference patterns. Naturally, for SU($N > 2$) fermions such interferograms prove to not be a good measure as explained above. Another interesting avenue to explore is the study of the phase diagrams of SU($N$) attractive fermions. The capability of SU($N$) fermions to form bound states of different types and natures, gives rise to rich and exotic phases [59]. This issue will be addressed in an upcoming work.

We note that, by monitoring the number of dislocations at weak interactions (repulsive or attractive), we can gain knowledge on $N_p/N$ –Figs. 6 and 7 - this feature provides the generalization of [55] to $N$-component fermions. Going to the strongly interacting regimes, the number and configuration of the dislocations changes, reflecting the persistent current fractionalization –Figs. 8 and 9. To be specific, there are $\lceil \frac{N_p}{2} \rceil + 1$ interference patterns with various dislocation numbers and orientation. This feature implies that, for repulsive interactions, we can access the number of particles $N_p$; for attractive interaction we can access on the number of components $N$ (see [56] for characterization of SU($N$) systems through neural networks).

In summary, we have shown how one can characterize SU($N$) correlated matter-wave through homodyne and self-heterodyne interference patterns. We believe that our findings are well within the current state-of-the-art of the field and can be experimentally traced. For the repulsive case, the experimental analysis could be carried out through the quantum gas microscopy [72–75].

## Acknowledgements

We acknowledge fruitful discussions with K. C. Wright.

## A    Homodyne detection protocol

### A.1    Analytical derivation of the momentum distribution at zero interactions

In cold atoms systems, the persistent current is experimentally observed through time-of-flight (TOF) imaging. This entails looking at the momentum distribution of the gas, which is one of the few observables that can be experimentally probed [76]. Here, we will give the derivation

for the momentum distribution.

Starting from the expression for the one-body correlator $n(\mathbf{r}, \mathbf{r}', t)$ defined as

$$\langle n(\mathbf{r}, \mathbf{r}')\rangle = \langle \Psi^\dagger(\mathbf{r})\Psi(\mathbf{r}')\rangle, \tag{A.1}$$

where $\mathbf{r}$ is position, and $\Psi^\dagger(\mathbf{r})$ and $\Psi(\mathbf{r})$ are the fermionic creation and annihilation field operators. Expanding the field operators in the basis set of the single-band Wannier functions $w_j(\mathbf{r})$ such that $\Psi(\mathbf{r}) = \sum_j^L w_j(\mathbf{r} - \mathbf{r}_j)c_j$, the two-point correlator has the following expression

$$\langle n(\mathbf{r}, \mathbf{r}')\rangle = \sum_{j,l}^L w_l^*(\mathbf{r} - \mathbf{r}_l)w_j(\mathbf{r}' - \mathbf{r}_j')\langle c_l^\dagger c_j\rangle, \tag{A.2}$$

with $w_j(\mathbf{r} - \mathbf{r}_j)$ being the Wannier function localised at site $j$ with position $r_j$ and $L$ being the number of lattice sites. If we consider the free expansion in time $t$ of the particle density $n(\mathbf{r}, \mathbf{r}', t)$, it is still defined as in Equation (A.2), but the time dependence is encoded in the expansion of the Wannier function $w_j(\mathbf{r}, t)$ that reads

$$w_j(\mathbf{r} - \mathbf{r}_j, t) = \frac{1}{\sqrt{\pi}} \frac{\eta_j}{\eta_j^2 + \iota\omega_0 t} \exp\left\{-\frac{(\mathbf{r} - \mathbf{r}_j)^2}{2(\eta_j^2 + \iota\omega_0 t)}\right\}, \tag{A.3}$$

where $\eta_j$ is the width of the center at the $j$-th site and $\omega_0 = \frac{\hbar}{m}$ with $\hbar$ and $m$ denoting Planck's constant and the particles' mass respectively, both of which are set to 1 in this calculation. Note that we have taken the zeroth order approximation of the Wannier function and the harmonic approximation. By letting the density distribution to expand for large values of time, one obtains the momentum distribution $n(\mathbf{k})$. The momentum distribution is defined as the Fourier transform of the one-body correlator $n(\mathbf{r}, \mathbf{r}')$,

$$\langle n(\mathbf{k})\rangle = \int e^{\iota\mathbf{k}(\mathbf{r} - \mathbf{r}')}\langle \Psi^\dagger(\mathbf{r})\Psi(\mathbf{r}')\rangle d\mathbf{r} d\mathbf{r}', \tag{A.4}$$

where $\mathbf{k}$ is the momentum. One can verify that $\lim_{t\to\infty}\langle n(\mathbf{r}, \mathbf{r}', t)\rangle \approx \langle n(\mathbf{k})\rangle$, by taking the limit $t \to \infty$ of Equation (A.3) and performing a Taylor expansion.

Substituting the expression for the field operators into Equation (A.4), the expression for $n(\mathbf{k})$ reads

$$\langle n(\mathbf{k})\rangle = \int e^{\iota\mathbf{k}(\mathbf{r} - \mathbf{r}')} \sum_{j,l}^L [w_l^*(\mathbf{r} - \mathbf{r}_l)w_j(\mathbf{r}' - \mathbf{r}_j')\langle c_l^\dagger c_j\rangle] d\mathbf{r} d\mathbf{r}'. \tag{A.5}$$

Utilising the change of variables $\mathbf{R} = \mathbf{r} - \mathbf{r}_j$ and $\mathbf{R}' = \mathbf{r}' - \mathbf{r}_l'$, we arrive to

$$\langle n(\mathbf{k})\rangle = \int e^{\iota\mathbf{k}(\mathbf{R} - \mathbf{R}')} \sum_{j,l}^L e^{\iota\mathbf{k}(\mathbf{r}_l - \mathbf{r}_j')}[w_l^*(\mathbf{R})w_j(\mathbf{R}')\langle c_l^\dagger c_j\rangle] d\mathbf{R} d\mathbf{R}', \tag{A.6}$$

which by making use of the fact that $w(\mathbf{k}) = \int w(\mathbf{R})e^{\iota\mathbf{k}\cdot\mathbf{R}}d\mathbf{R}$, can be further simplified to give

$$\langle n(\mathbf{k})\rangle = |w(\mathbf{k})|^2 \sum_{j,l}^L e^{\iota\mathbf{k}(\mathbf{r}_l - \mathbf{r}_j')}\langle c_l^\dagger c_j\rangle, \tag{A.7}$$

with $w(\mathbf{k})$ being the Fourier transform of $w(\mathbf{R})$. Finally, we write our ring configuration explicitly as

$$\langle n(\mathbf{k})\rangle = |w(\mathbf{k})|^2 \sum_{j,l}^L e^{\iota\left[k_x r\left(\cos\left(\frac{2\pi l}{L}\right) - \cos\left(\frac{2\pi j}{L}\right)\right) + k_y r\left(\sin\left(\frac{2\pi l}{L}\right) - \sin\left(\frac{2\pi j}{L}\right)\right)\right]}\langle c_l^\dagger c_j\rangle, \tag{A.8}$$

where the momentum vector can be written as $\mathbf{k} = (k_x, k_y)$ and polar coordinates were introduced $\mathbf{r} = (r\cos\theta, r\sin\theta)$, with $\theta = \frac{2\pi}{L}l$. We introduce the creation operator and its Fourier transform

$$c_l^\dagger = \frac{1}{\sqrt{L}} \sum_k^L e^{-\imath kl} c_k^\dagger,$$  (A.9)

as well as the Fourier transform of its counterpart the annihilation operator

$$c_j = \frac{1}{\sqrt{L}} \sum_{k'}^L e^{\imath k' j} c_{k'}.$$  (A.10)

The one-body correlator $\langle c_l^\dagger c_j \rangle$ in Fourier space reads

$$\langle c_l^\dagger c_j \rangle = \frac{1}{L} \sum_{k,k'}^L e^{-\imath kl + \imath k' j} \langle c_k^\dagger c_{k'} \rangle.$$  (A.11)

Furthermore, at zero interactions we have that $\langle c_k^\dagger c_{k'} \rangle = \delta_{k,k'}$. Consequently, the expression for the momentum distribution reads

$$\langle n(\mathbf{k}) \rangle = \frac{1}{L} |w(\mathbf{k})|^2 \sum_{j,l}^L e^{\imath \left[ k_x r \left( \cos\left(\frac{2\pi l}{L}\right) - \cos\left(\frac{2\pi j}{L}\right) \right) + k_y r \left( \sin\left(\frac{2\pi l}{L}\right) - \sin\left(\frac{2\pi j}{L}\right) \right) \right]} \sum_{\{q\}} e^{-\frac{2\pi\imath}{L}(l-j)q},$$  (A.12)

where we made use of the fact that for free fermions $k = \frac{2\pi}{L}q$ with $q$ being the quantum number labeling the Fermi sphere levels. At this point, by setting $k_x = |\mathbf{k}|\sin\phi$ and $k_y = |\mathbf{k}|\cos\phi$ we have that

$$\langle n(\mathbf{k}) \rangle \propto \frac{1}{L} \sum_{\{q\}} \left| \sum_l^L e^{\imath r(B\sin\phi\cos\theta_l + B\cos\phi\sin\theta_l)} e^{-\frac{2\imath\pi l}{L}q} \right|^2,$$  (A.13)

where $B = \sqrt{2|\mathbf{k}|^2}$. Using the trigonometric identity $\sin(A+B) = \sin A \cos B + \cos A \sin B$, the expression is simplified even further and reads

$$\langle n(\mathbf{k}) \rangle \propto \frac{1}{L} \sum_{\{q\}} \left| \sum_l^L e^{\imath r B \sin(\phi + \theta_l)} e^{-\frac{2\imath\pi l}{L}q} \right|^2.$$  (A.14)

The expression in Equation (A.14) is an $q$-th order Bessel function of the first kind [55]

$$J_q(x) = \frac{1}{2\pi} \int_{-\pi}^{\pi} e^{\imath(x\sin\tau - q\tau)},$$  (A.15)

where $x = rB$ and $\tau = \frac{2\pi l}{L}$. It is important to stress here that the replacing the sum by an integral is only an approximation, which becomes valid in the thermodynamic limit. As a result, by setting

$$J_q(B) \approx \sum_l^L e^{\imath r B \sin(\phi + \theta_l)} e^{-\frac{2\imath\pi l}{L}q},$$  (A.16)

we have that the momentum distribution reads as

$$\langle n(\mathbf{k}) \rangle \propto \frac{1}{L} \sum_{\{q\}} |J_q(\mathbf{k})|^2.$$  (A.17)

This enables us to study the momentum distribution analytically, by considering it as a summation of different Bessel functions as was carried out in [55] and generalized to SU($N$) in the main text. It is important to note that the Equation (A.17) only holds at zero interactions. In the case of interacting particles $\langle c_k^\dagger c_{k'} \rangle \neq \delta_{k,k'}$. As such, we no longer have an analytical expression for the momentum distribution and different behaviours are observed in the interacting regimes as reported in this paper.

## A.2 Emergence of a hole in the momentum distribution

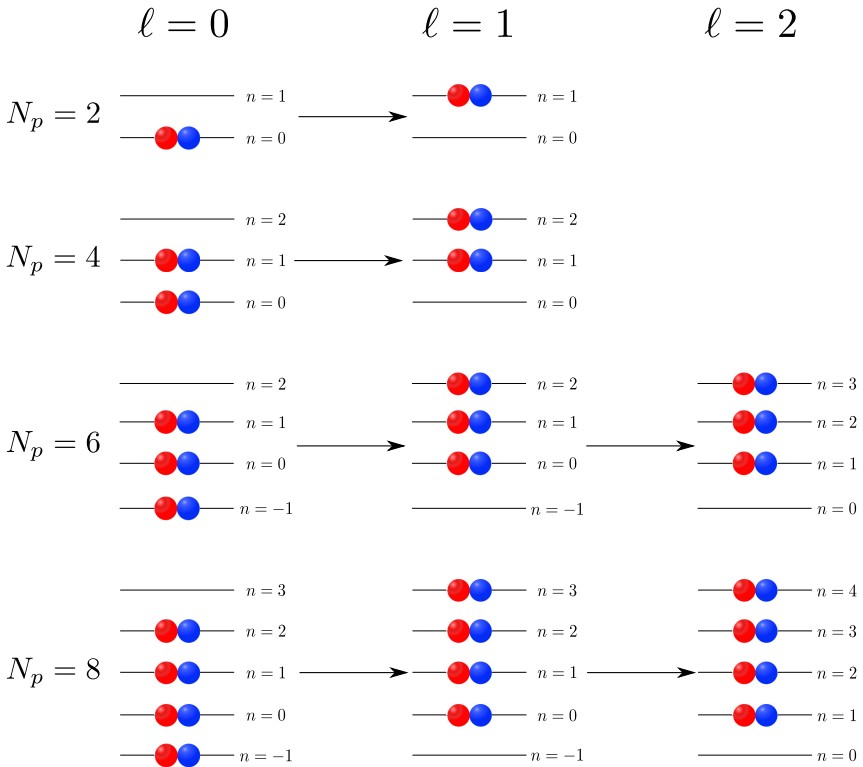

Figure 10: SU(2) particle occupation energy levels for $N_p = 2, 4, 6, 8$ particles and their displacement with angular momentum quantization $\ell$ at zero interaction. It is clear that with increasing $N_p$ one needs a higher value of $\ell$ such that the particles do not occupy the $n = 0$ level. Note that there are two different parity cases corresponding to $N_p = (2m+1)N$ and $N_p = (2m)N$ with $m$ being an integer number.

For a hole to open, one requires that the momentum distribution collapses to zero at the origin. This can only occur when the Fermi sphere is displaced by $\lceil \frac{N_p}{2N} \rceil$ and there are no particles occupying the $n = 0$ level. The distribution of the particles needs to be such that it is symmetrical around the $n = 0$ level.

*Non-interacting:* Let us take a look at Fig. 10 where we consider the occupation of SU(2) particles at zero interaction. In the case of $N_p = 2$ and $N_p = 4$, we see that as we increase the effective magnetic flux $\phi$ and pass from the first parabola with angular momentum quantization $\ell = 0$ to the second one with $\ell = 1$ (see Fig. 11), the Fermi sphere is displaced such that there are no particles occupying the $n = 0$ level. In this case, $\ell = 1$ corresponds to the angular momentum $\ell_H$ for a hole to open up with $\phi_H = 0.5$ ($\phi_H = 1.0$) for $N_p = 2$ ($N_p = 4$). The value of $\phi_H$ corresponds to the flux value where we traverse to the energy parabola with angular momentum $\ell_H$. In the case of $N_p = 6$ and $N_p = 8$, in contrast with the $N_p = 2$ and $N_p = 4$, one

needs to go the third parabola with $\ell = 3$ to clear the $n = 0$ level. As such, the opening of the hole in the momentum distribution is delayed since a higher value of $\phi$ is required.

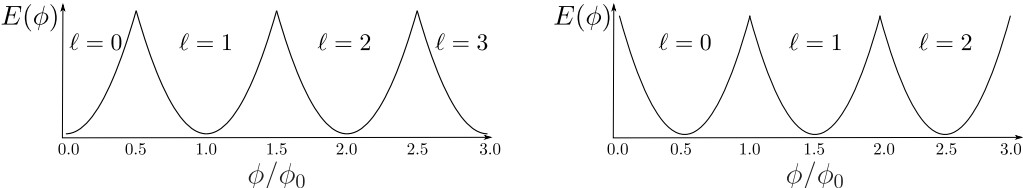

Figure 11: Energy as a function of the effective magnetic flux $\phi$, denoted by $E$ and $\phi$ respectively, for systems with $N_p = (2m + 1)N$ (left) and $N_p = (2m)N$ (right). As one crosses from one parabola to the next with increasing $\phi$, the angular momentum quantum number $\ell$ increases. The difference between the left and right panels stems from the parity of the system which is diamagnetic and paramagnetic respectively depending on whether the ground-state energy increases or decreases with flux $\phi$ [43,60]. The degeneracy point, which is the point where two parabolas cross, is at (half-odd) integer values for (diamagnetic) paramagnetic systems.

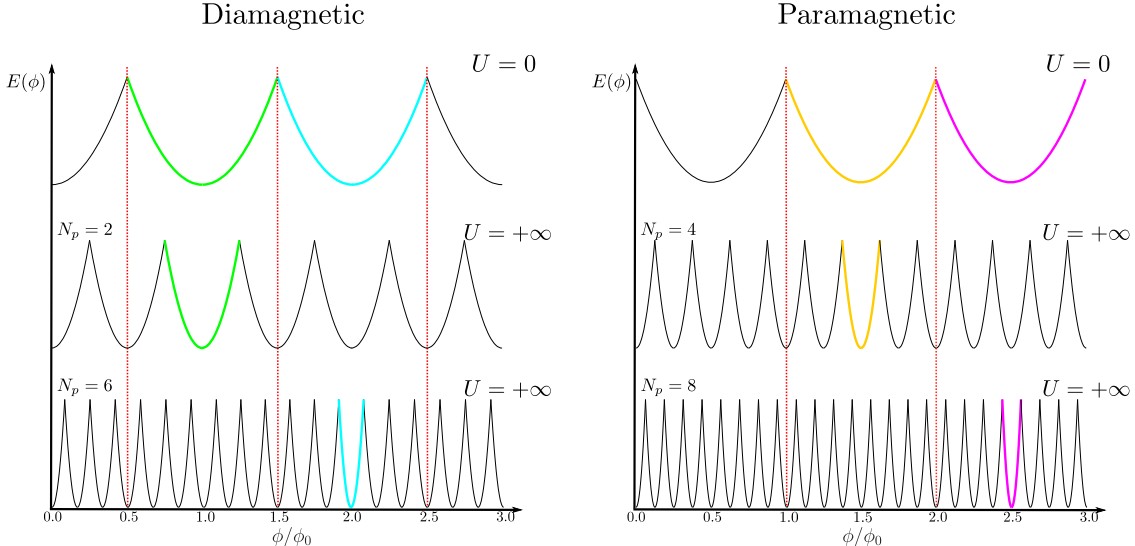

Figure 12: Schematic figure of the energy $E$ against effective magnetic flux $\phi$ for diamagnetic (left) and paramagnetic (right) cases. Top panel depicts the diamagnetic and paramagnetic cases at zero interaction for SU(2) fermions (holds for any $N$). The middle panel shows the system for 2 and 4 particles with infinite repulsive interaction $U$. Comparing this panel with the one at $U = 0$, we see that the hole for $N_p = 2$ (green) and $N_p = 4$ (yellow) is delayed on going to infinite repulsion. Likewise for $N_p = 6$ (cyan) and $N_p = 8$ (magenta), we also observe a delay in addition to the one that is observed at $U = 0$ (see Fig. (11)).

*Repulsive:* Turning our attention to the infinitely repulsive case, we observe a depression instead of a hole. If we consider $N_p = 2$ with SU(2) symmetry with infinite repulsion, we see the depression at a higher value of the flux than the non-interacting case. Indeed, we find the depression with a 'delay' of $\frac{1}{2N_p}$. Going to the four particle case, the depression is delayed by $\frac{3}{2N_p}$ (reduced period of one of the parabolas is $1/N_p$). Considering more cases, we find that for a depression to appear for a given $N_p$ with infinite repulsion, $\phi_D = \phi_H + \frac{N_p - 1}{2N_p}$.

The depression is tracked through the second derivative of the momentum distribution $n(k_x, k_y)$ at $k_x = k_y = 0$, defined as $\frac{\partial^2 n(k_x, 0)}{\partial k_x^2}\big|_{k_x=0}$. By noting how the derivative changes from negative to positive, which corresponds to a peak and depression respectively, we are able to observe at what values of the flux $\phi$ a depression appears for a given interaction $U$. In turn, the second derivative gives us insight into the fractionalization of the persistent current by monitoring the 'delay' of the depression.

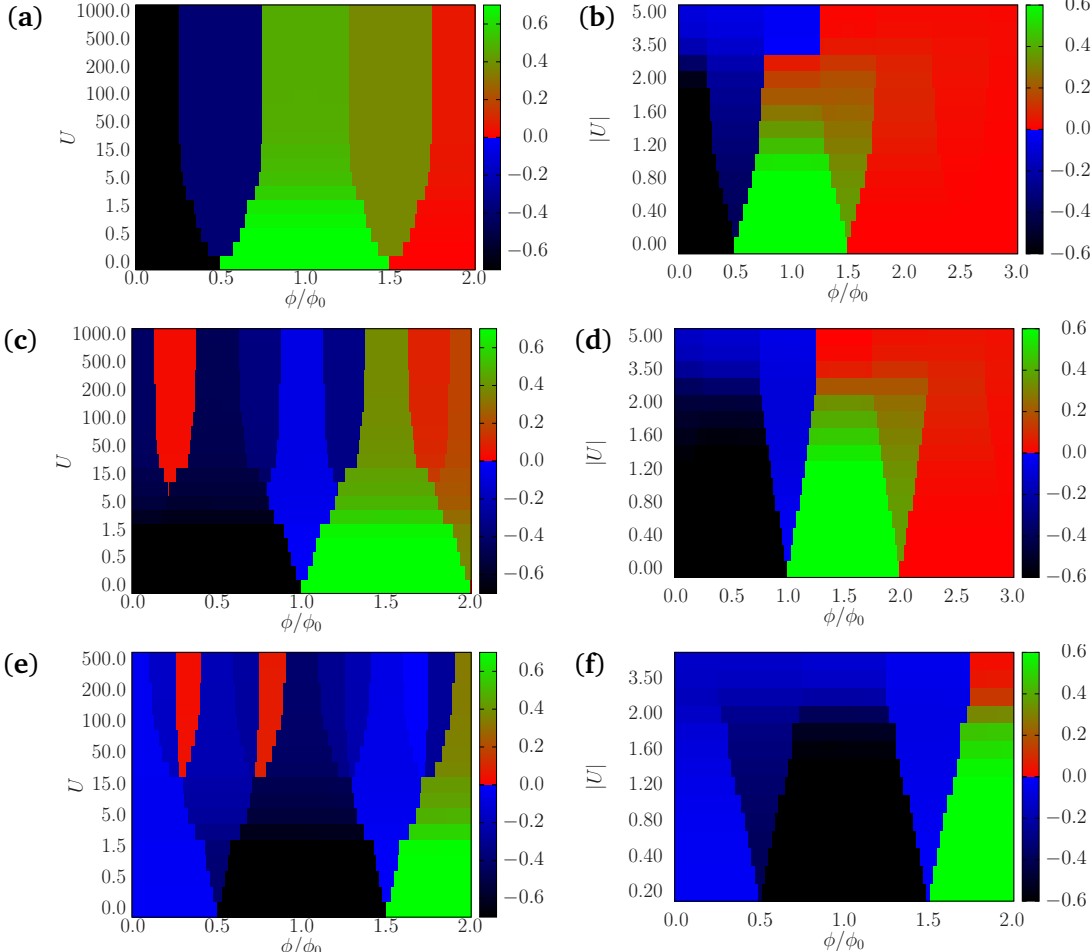

Figure 13: Second derivative of the momentum distribution $\frac{\partial^2 n(k_x, 0))}{\partial k_x^2}\big|_{k_x=0}$ evaluated at $k_x = k_y = 0$, as a function of the effective magnetic flux $\phi$ and interaction $U$. The left and right panels correspond to the repulsive and attractive regimes respectively for $N_p = 2$ (top), $N_p = 4$ (middle) and $N_p = 6$ (bottom) SU(2) fermions. In both regimes, we observe that (i) as the number of particles increase the depression is delayed to higher flux values; (ii) there is an extra delay for stronger interactions. However, a notable difference is that in the attractive case the depression is smoothed out with increasing interactions. Results were obtained with exact diagonalization for $L = 15$ sites. Note that the $y$-axis is not linear in the values of the interaction.

Fig. 13 corresponds to systems with repulsive interactions. As previously discussed, the value at which the depression is observed (transition from blue to green) appears at a larger $\phi$ with increasing $U$. As $U \to \infty$, the system attains complete fractionalization, which is reflected from the fixed value of the flux, corresponding to $\phi_D$. A peculiar phenomenon in Fig. 13 (**b**)

and (**c**) is the alternation between peak (blue) and depression (red) at flux values preceding $\phi_H$. We would like to point out that the red areas are small in value that would correspond to plateaus in an experimental setting.

*Attraction:* For infinitely attractive interactions, the energy also fractionalizes with a reduced period of $\phi_0/N$ irrespective of the number of particles. For intermediate interactions, there is a momentum distribution depression appearing at $\phi_D = \phi_H + \frac{N-1}{2N}$ –Fig. 4. Indeed, just like in its repulsive counterpart, the depression experiences a two-fold 'delay'. However, in this case the 'delay' depends on both the particle number (through $\phi_H$) and on the number of components (second term).

Once the system fractionalizes fully, indicating the formation of the $N$-body bound state, the depression appears at $\phi_D$. The depression is small due to the reduced coherence. If the interaction keeps increasing, one will no longer find a depression at $\phi_D$. Indeed, larger $\phi$ values are required for a depression to appear. Unlike the repulsive case, there is no alternation between depression and peak.

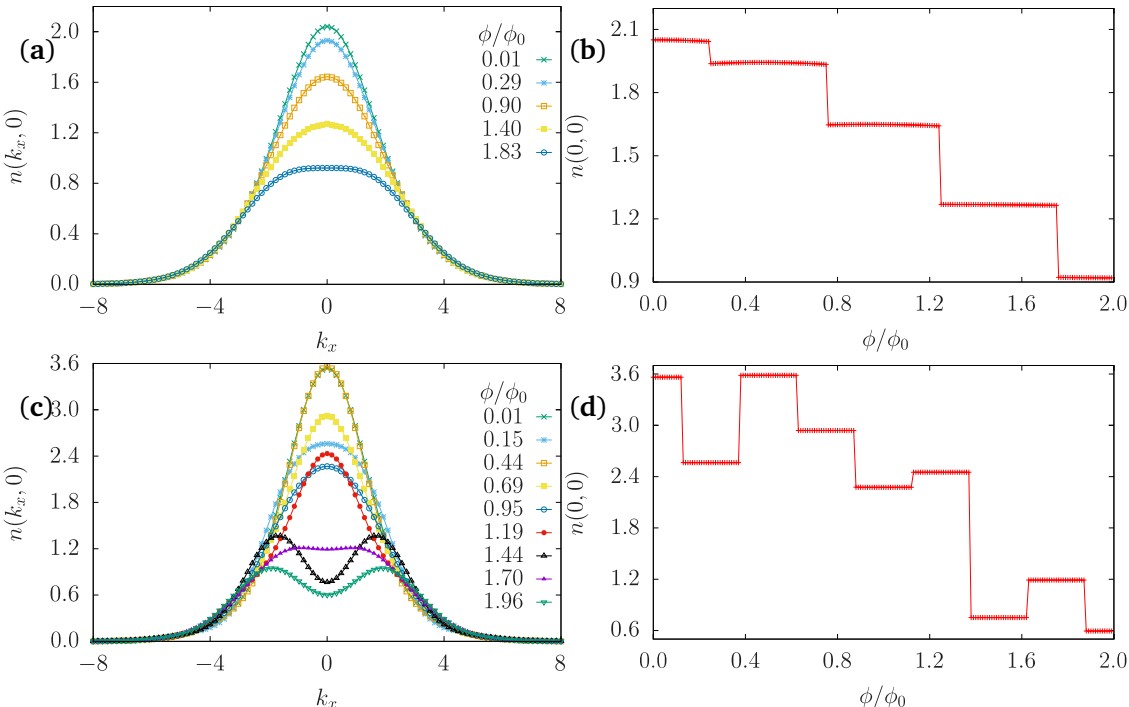

Figure 14: Left panel: Cross-section of the momentum distribution $n(k_x, 0)$ for $U = -5$ (top) and $U = 10,000$ (bottom) as a function of the effective magnetic flux $\phi$. On the right panels, there is the corresponding momentum distribution $n(0,0)$ for $k_x = k_y = 0$. Results were obtained with exact diagonalization for $N_p = 4$, $N = 2$ for $L = 15$.

The peculiar behaviour observed in Fig. 13 (**b**) and (**c**) can be understood through the cross-section of the momentum distribution depicted in Fig. 14 (**c**). Indeed, we find that the height of the momentum distribution peak/depression varies with $\phi$. This phenomenon is more clear if one looks at the momentum distribution $n(0,0)$ at $k_x = k_y = 0$, which shows the non-monotonous behaviour in the momentum distribution. On the other hand, the behaviour is monotonic for attractive interactions –Figs. 14 (**a**) and (**b**). The non-monotonous behaviour observed at infinite repulsion is the reason why the variance of the momentum distribution does not give good diagnostic tool in this regime. This is in sharp contrast with its attractive

counterpart.

## A.3 Expansion of the density distribution

The momentum distribution is obtained by releasing the atoms from the trap and observing the particle density distribution after a long expansion time. Initially at time $t = 0$, one observes the Wannier function localized on each site. Then on going to intermediate times, the ring expands to a characteristic hole with protruding spirals giving rise to a peculiar shape resembling a 'shuriken'–Figure 15.

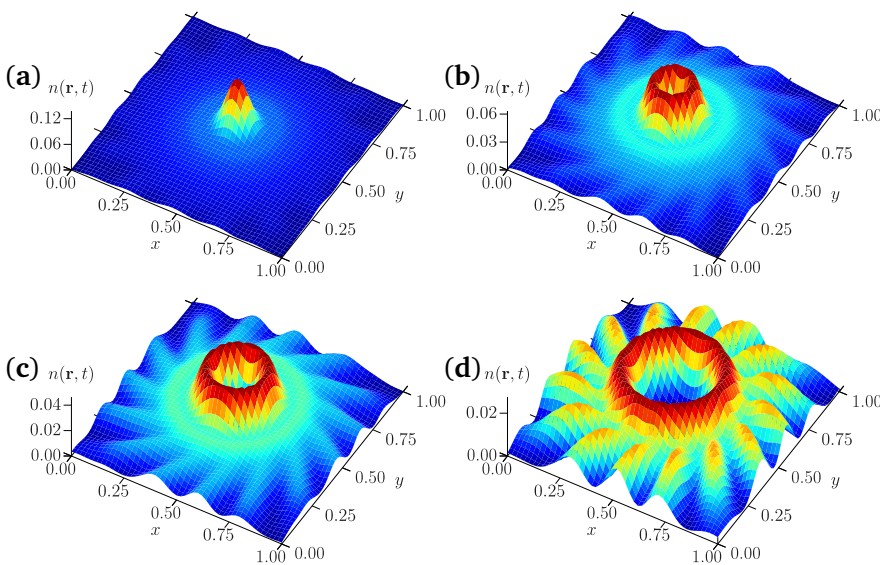

Figure 15: Density distribution $n(\mathbf{r}, t)$ at an intermediate time $t = 3$, for various values of the flux $\phi$. On increasing the flux, we go from a sharply peaked Gaussian (leftmost panel) in the middle, to a characteristic hole with spirals radiating from it. Eventually, as the size of the hole increases, the intensity from the spirals increases (rightmost panel), resembling a 'shuriken'. The results were calculated with exact diagonalization for $N_p = 4$ with $N = 2$ and $L = 15$ at $U = 0$ for $\phi/\phi_0 = 0, 1, 2, 4$.

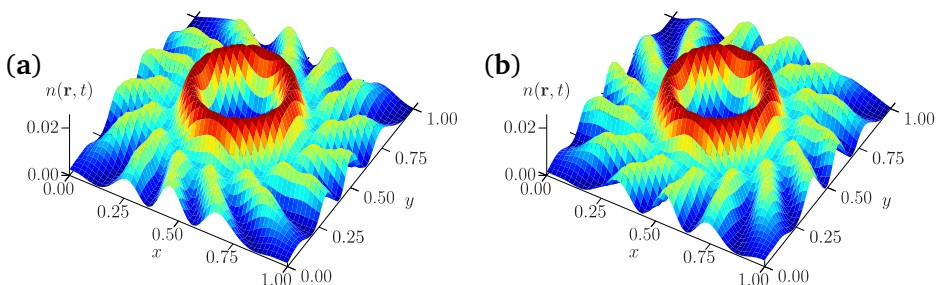

Figure 16: Density distribution $n(\mathbf{r}, t)$ at an intermediate time $t = 3$ evaluated at $\phi/\phi_0 = 4$ and $\phi/\phi_0 = -4$. The right (left) panel corresponds to $\phi = 4$ ($\phi = -4$) and the direction of the protruding spirals is clockwise (anti-clockwise). The direction of the rotation by the artificial gauge field is reflected in the quantum shuriken.The results were calculated with exact diagonalization for $N_p = 3$ with $N = 3$ and $L = 15$ at $U = 0$.

The direction of the spirals, be they clockwise or anti-clockwise gives an indication about the directional flow of the current –see Fig. 16. Eventually at longer times, one recovers the characteristic momentum distribution.

## B  Self-heterodyne detection protocol

Here, we consider the density-density correlator $G(\mathbf{r}, \mathbf{r}', t)$ for an expanding ring and an additional site in the center at a fixed time $t$. The two-body correlator is defined in the following way

$$G(\mathbf{r}, \mathbf{r}', t) = \sum_{\alpha,\beta}^{N} \langle n_\alpha(\mathbf{r}, t) n_\beta(\mathbf{r}', t) \rangle. \tag{B.1}$$

The density operator is defined as $n(\mathbf{r}, t) = \psi^\dagger(\mathbf{r}, t)\psi(\mathbf{r}, t)$ where $\psi^\dagger = (\psi_R^\dagger + \psi_C^\dagger)$ being the field operator of the whole system of the ring and the center, denoted by $R$ and $C$ respectively. Initially, the ring and the center are decoupled until they are released from their confinement potential. Thus, at time $t = 0$ the ground-state can be seen as a product state $|\phi\rangle = |\phi\rangle_R \otimes |\phi\rangle_C$.

Assuming free expansion for $t \geq 0$, the number of terms in the $G(\mathbf{r}, \mathbf{r}', t)$ can be significantly reduced. Firstly, terms consisting of an odd number of creation or annihilation operators have a null expectation value due to particle conservation. Likewise, terms where either both creation or annihilation operators act on one system also vanish. As such, the only surviving terms are those comprised of an equal number of creation-annihilation pairs, one acting on the ring and another on the center. Consequently, the expression for the density-density correlator reads

$$
\begin{aligned}
\sum_{\alpha,\beta}^{N} \langle n_\alpha(\mathbf{r}, t) n_\beta(\mathbf{r}', t) \rangle = {}& \sum_{\alpha,\beta}^{N} \langle n_\alpha(\mathbf{r}, t) n_\beta(\mathbf{r}', t) \rangle_R + \langle n_\alpha(\mathbf{r}, t) n_\beta(\mathbf{r}', t) \rangle_C \\
& + \sum_{\alpha,\beta}^{N} \langle n_\alpha(\mathbf{r}, t) \rangle_R \langle n_\beta(\mathbf{r}', t) \rangle_C + \langle n_\beta(\mathbf{r}, t) \rangle_C \langle n_\alpha(\mathbf{r}', t) \rangle_R \\
& + \sum_{\alpha,\beta}^{N} \langle \phi_C | \psi_{C,\alpha}^\dagger(\mathbf{r}) \psi_{C,\beta}(\mathbf{r}') | \phi_C \rangle [\delta(\mathbf{r} - \mathbf{r}') - \langle \phi_R | \psi_{R,\beta}^\dagger(\mathbf{r}') \psi_{R,\alpha}(\mathbf{r}) | \phi_R \rangle] \\
& + \sum_{\alpha,\beta}^{N} [\delta(\mathbf{r} - \mathbf{r}') - \langle \phi_C | \psi_{C,\alpha}^\dagger(\mathbf{r}) \psi_{C,\beta}(\mathbf{r}') | \phi_C \rangle] \langle \phi_R | \psi_{R,\beta}^\dagger(\mathbf{r}') \psi_{R,\alpha}(\mathbf{r}) | \phi_R \rangle.
\end{aligned}
\tag{B.2}
$$

The first four terms in Equation (B.2) do not give rise to any interference patterns. Indeed, it is the cross-terms between the ring and the center (last two terms) that give rise to interference. Therefore, taking these two terms and decomposing into the Wannier states yields

$$G_{R,C} = \sum_{\alpha,\beta}^{N} \sum_{j,l=1}^{L} I_{jl}(\mathbf{r}, \mathbf{r}', t) \big[ N_0 (\delta_{jl} - \langle \phi_R | c_{l,\alpha}^\dagger c_{j,\alpha} | \phi_R \rangle) + (1 - N_0) \langle \phi_R | c_{l,\alpha}^\dagger c_{j,\alpha} | \phi_R \rangle \big], \tag{B.3}$$

which is the interference of the Wannier function where

$$I_{jl}(\mathbf{r}, \mathbf{r}', t) = w_c(\mathbf{r}', t) w_c^*(\mathbf{r}, t) w_l^*(\mathbf{r}' - \mathbf{r}'_l, t) w_j(\mathbf{r} - \mathbf{r}_j, t),$$

and $N_0 = \langle \phi_C | c_{0,\beta}^\dagger c_{0,\beta} | \phi_C \rangle$ defines the expectation value of the number operator center, which in the current protocol is always equal to one. Consequently, the second term in Equation (B.3)

does not contribute to the interference pattern. Note that one of the summations over the number of components is removed due to the Kronecker delta $\delta_{\alpha\beta}$ that arises due to the colour conservation nature of the Hamiltonian describing the system. To enhance the visibility of the spirals, we neglect the Kronecker delta in the first term of Equation (B.3).

## B.1 Repulsive interactions

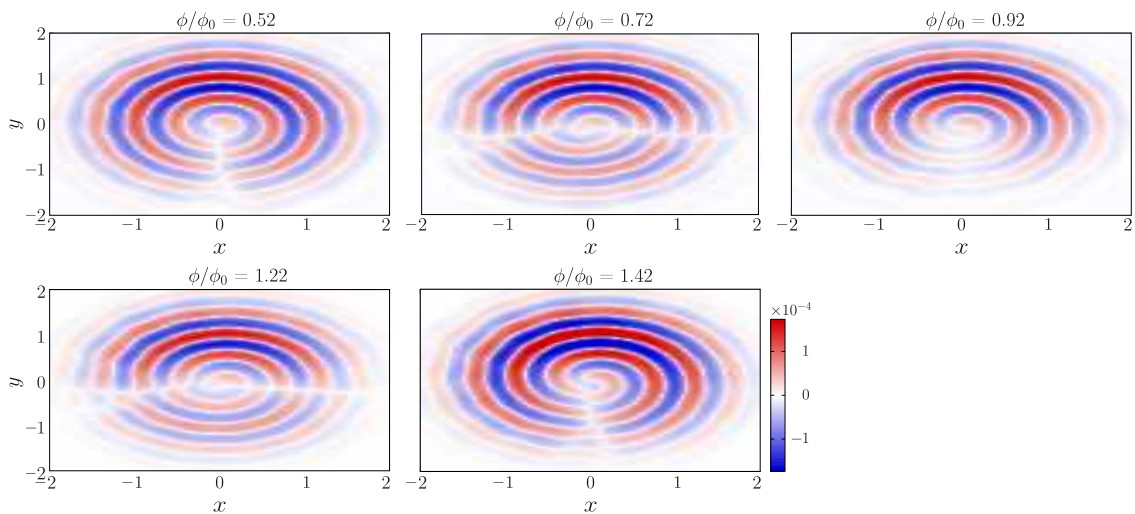

Figure 17: The interference $G_{\mathrm{R,C}}$ between ring and center for $N_p = 4$ particles with SU(2) symmetry, is shown as a function of the effective magnetic flux $\phi$ at $U = 5000$ and short time $t = 0.026$. All correlators are evaluated with exact diagonalization for $L = 15$ by setting $\mathbf{r}' = (0,R)$ and radius $R = 1$. The color bar is non-linear by setting $\mathrm{sgn}(G_{\mathrm{R,C}})\sqrt{|G_{\mathrm{R,C}}|}$.

The characteristic spirals are also observed in interferograms at infinite repulsion. We find that multiple spirals appear at different values of the flux, that for the sake of simplicity we are going to distinguish as A and B. The emergence of the multiple spirals can be clearly seen by looking at Fig. 17, which depicts the interfergorams at infinite repulsion on going from $\ell = 0$ to $\ell = 1$. At $\phi = 0.92$, we observe spiral B that disappears on going to the next parabola with $\phi = 1.22$. The last panel shows the appearance of spiral A where $\phi$ exceeds $\phi_{S_A}$. This spiral persists on going to the next parabolas.

Indeed, their emergence is not as clear cut as the zero interaction case. The appearance of spiral A coincides with the depression of the momentum distribution, occurring at $\phi_{S_A} = \phi_H + \frac{N_p - 1}{2N_p}$. Spiral A gains more arms as the angular momentum is increased, as in the zero interaction case. However, spiral B appears when the angular momentum of the system corresponds to $\ell_H$, the angular momentum at which a hole opens up at $U = 0$. In particular, it appears at $\phi_{S_B} = \phi_H + \frac{1}{2N_p} + \Delta$ ($\phi_{S_B} = \phi_H - \frac{1}{2N_p}$) for a diamagnetic and paramagnetic system respectively, with $\Delta = -\frac{1}{2N_p}$ for an odd number of particles. and zero otherwise This spiral only appears for the period of that parabola, which corresponds to $\frac{1}{N_p}$. The additional term $\frac{1}{2N_p}$ takes into account the change in the profile of the energy brought on by the level crossings – see Fig. 12. In the special case of $N_p = 2$ SU(2) particles, where the second term of $\phi_S$ reads as $\frac{1}{2N_p}$, spiral A and B are one and the same.

It is important to note that due to the large number of dislocations present in the system with infinite repulsive interactions, it becomes harder to deduce the nature of the spiral.

## B.2 Attractive interactions

As previously mentioned, the fractionalization in attractive fermionic systems depends only on the number of components. For of SU(2) systems, the fractionalization is such that we go from one parabola at $U = 0$ to three piece-wise parabolic segments as $U \to -\infty$. In line with what is observed for repulsive systems, the level crossings that occur during fractionalization can be observed in the self-heterodyne interferograms.

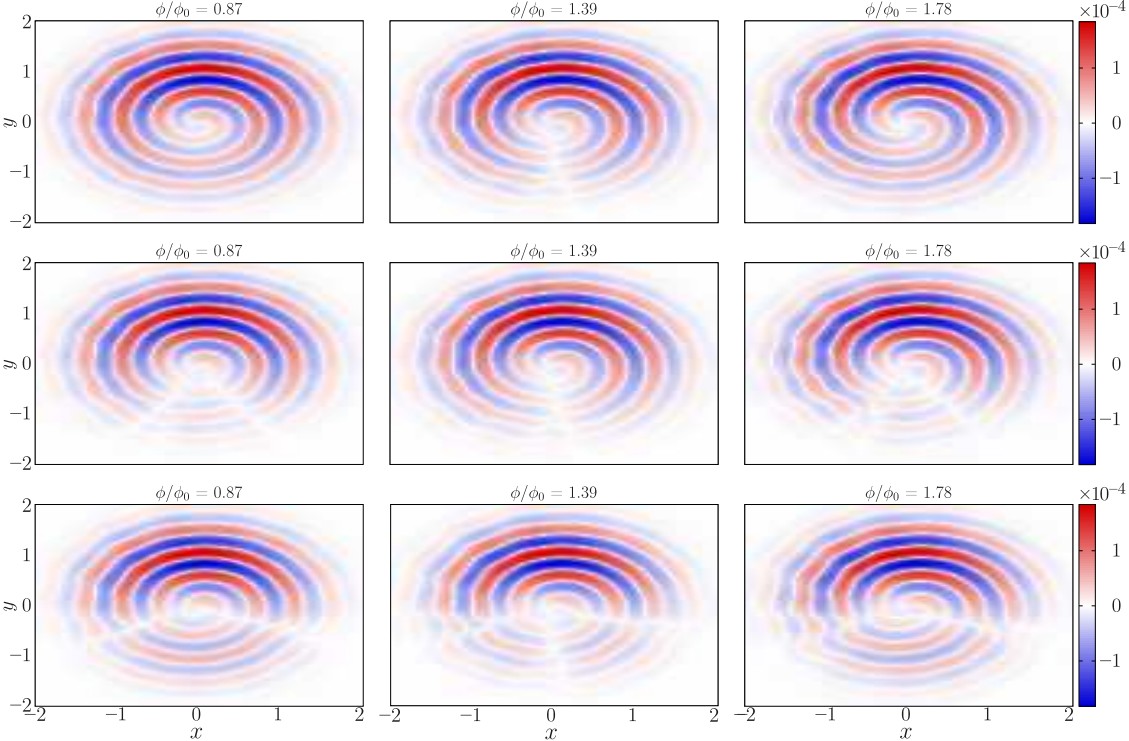

Figure 18: The interference $G_{\text{R,C}}$ between ring and center for $N_p = 2$ (top) $N_p = 4$ (middle) and $N_p = 6$ (bottom) particles with SU(2) symmetry, is shown as a function of the effective magnetic flux $\phi$ at $U = -3$ and short time $t = 0.025$. All correlators are evaluated with exact diagonalization for $L = 15$ by setting $\mathbf{r}' = (0, R)$ and radius $R = 1$. The color bar is non-linear by setting $\text{sgn}(G_{\text{R,C}})\sqrt{|G_{\text{R,C}}|}$.

Such behaviour can be clearly seen in Fig. 18 that shows the interferograms for $N_p = 2, 4, 6$ SU(2) fermions with attractive interactions. In all three cases, we see the appearance of an extra dislocation for $\phi$ corresponding to a fractionalized parabola. In the case of $N_p = 4$, there are left and right panels corresponding to the fractionalized parabola as opposed to the $N_p = 2, 6$ cases due to a different parity at $U = 0$.

Lastly, we have that the emergence of the spiral is delayed by $\frac{N-1}{2N}$. It is important to note that unlike the repuslive case, here we do not observe the emergence of the second spiral (called spiral B in Sec. B.1).

In the intermediate attractive interaction regime, we can track the partial fractionalization of the persistent current through self-heterodyne interference patterns just like its repulsive counterpart –Fig. 19. As the interaction is increased, the number and shape of dislocations changes with the effective magnetic flux threading the system, reflecting the the reduced periodicity of the current. Additionally, the emergence of the spiral experiences an additional delay due

to the fractionalization of the system. However, as explained above, the delay is solely dependent on the number of components and does is equivalent for systems having different particle numbers.

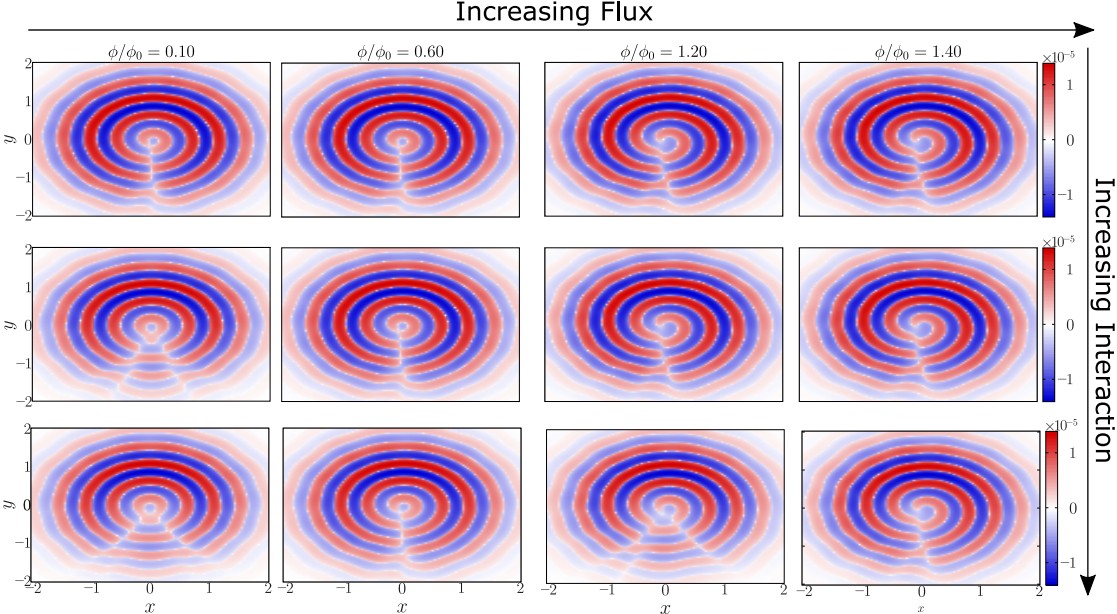

Figure 19: The interference $G_{R,C}$ between ring and center for $N_p = 4$ particles with SU(2) symmetry against the effective magnetic flux $\phi$ at short time $t = 0.0225$ as a function of the interaction $U$. In this figure, one can clearly see how the number and orientation of the dislocations change as one increases the interaction. All correlators are evaluated with DMRG for $L = 15$ and attractive $|U| = \{0, 1, 2\}$ (panels are in descending order) by setting $\mathbf{r}' = (0, R)$ and radius $R = 1$. The color bar is non-linear by setting $\text{sgn}(G_{R,C})|G_{R,C}|^{1/4}$.

### B.2.1 CSF configuration

In the main text, it was stated that SU($N$) fermions are capable of forming bound states having different types and natures. For instance, SU(3) symmetric fermions form two types of bound states: trions where all three colours are bounded; and colour superfluids (CSFs) having two of the colours in a pair with the other one remaining unpaired [59]. It has been demonstrated in an earlier work [44], that the persistent current is able to distinguish between these two bound states, which in turn can be read-out through the momentum distribution in the TOF expansion. Previously, we have explained that the self-heterodyne interference patterns are found to not be able to provide an observable to monitor the persistent current pattern, since a higher order correlator is probably required to capture the features of an $N$-body bound state. Indeed, three-body bound states of SU(3) fermions were not able to be analysed via self-heterodyne interferences. However, we find that CSFs being two-body bound states, can be analysed through observables obtained with the self-heterodyne protocol. In what follows, we provide a brief explanation on how to read-out the persistent currents of CSFs through self-heterodyne interferograms.

For CSF bound states to form in a canonical ensemble, which is the case considered here, one needs to break the SU(3) symmetry. Here, this is carried out by choosing asymmetric interactions between the colours that we denoted as $A$, $B$ and $C$ such that $|U_{AB}| \neq |U_{BC}| \neq |U_{AC}|$. For a CSF, we require that one interaction between the colours is significantly larger than the

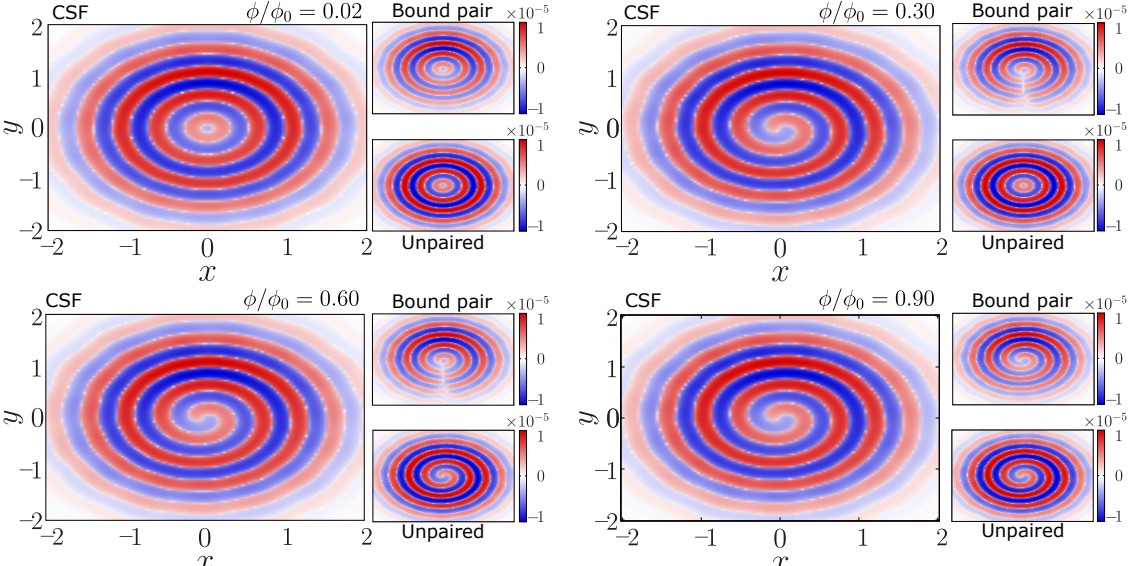

Figure 20: The interference $G_{R,C}$ between ring and center for $N_p = 3$ three-component fermions in a CSF configuration, is shown as a function of the effective magnetic flux $\phi$ at short time $t = 0.0225$. Main panels corresponds to the CSF, upper panel to the two-body bound state and bottom panel to the unpaired particles. All correlators are evaluated with exact DMRG for $L = 15$ by setting $\mathbf{r}' = (0, R)$ and radius $R = 1$. The CSF configurations is achieved by $|U_{AB}| = |U_{BC}| = 0.01$ and $|U_{AC}| = 3$. The color bar is non-linear by setting $\mathrm{sgn}(G_{R,C})|G_{R,C}|^{1/4}$.

other two such that for example $|U_{AC}| \gg |U_{AB}| = |U_{BC}|$. On account of the symmetry breaking, our analysis can be carried out by analysing the interference patterns of fermions of a given colour: i.e. analysis of the interferograms of the bound pair and unpaired is done separately instead of looking at the phase portrait of the CSF as a whole. Naturally, such an analysis relies on the capability to address fermions of different colours separately in experiments [29].

Figs. 20 depicts the interference patterns for $N_p = 3$ particles with asymmetric interactions such that the system is in a CSF configuration. The top row corresponds to the interference patterns of the CSF, where we observe the emergence of a spiral after displacing half the Fermi sphere. At a glance, it looks as if the interferograms of the CSF correspond to that of non-interacting particles. However, looking at the phase portraits of fermions in different colours paints a more interesting picture. For the bound pair, we observe that on going from one parabola to the other, the number and orientation of the dislocations change to account for the fractionalization. Additionally, the emergence of the spiral experiences an extra delay in the value of the flux that arises from the fractionalized parabolas. For the unpaired particle, as discussed in the main text, the interference patterns are the same as the free particle case as the interaction it experiences is very small. An interesting point worthy of mention is the lack of dislocations in the interference patterns of the free particle. It appears that on account of the SU(3) symmetry breaking, the Fermi spheres of the bound and free particles are essentially decoupled. Lastly, we point out that when considering the full interferogram, the dislocation that appears for the bound pair is not readily visible due to its reduced coherence, which is significantly smaller than that of the free particle. Such a statement is in line with the TOF distribution analysis carried out in [44]. The same observations hold when considering a large number of particles, with the added difference that there is an increased delay in the flux for the spiral to emerge and in the number of dislocations as can be readily observed from Fig. 21.

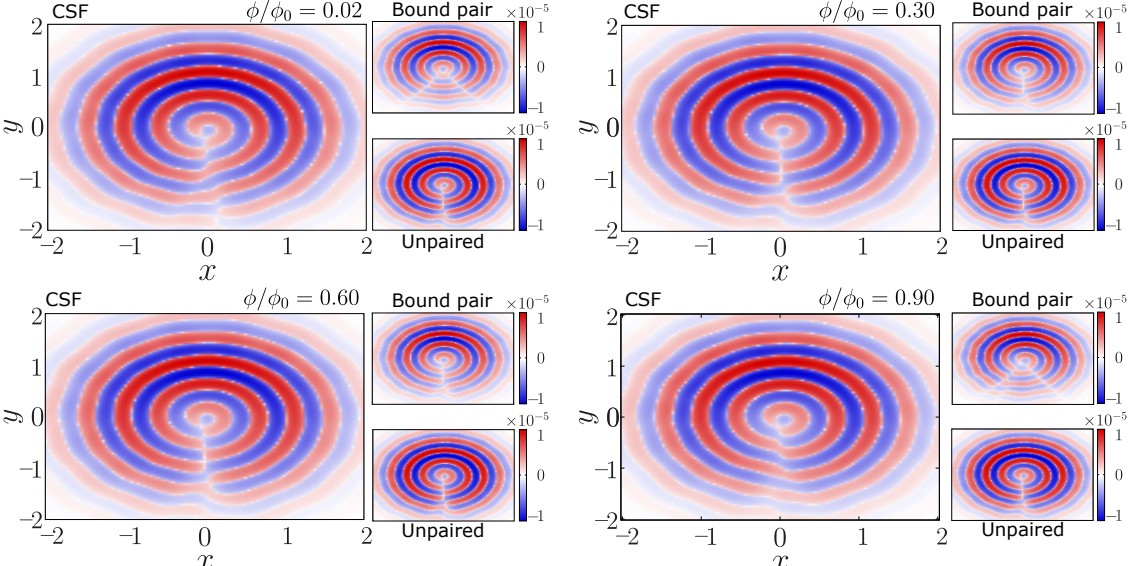

Figure 21: The interference $G_{\mathrm{R,C}}$ between ring and center for $N_p = 6$ three-component fermions in a CSF configuration, is shown as a function of the effective magnetic flux $\phi$ at short time $t = 0.0225$. Main panels corresponds to the CSF, upper panel to the two-body bound state and bottom panel to the unpaired particles. All correlators are evaluated with DMRG for $L = 15$ by setting $\mathbf{r}' = (0, R)$ and radius $R = 1$. The CSF configurations is achieved by $|U_{AB}| = |U_{BC}| = 0.01$ and $|U_{AC}| = 3$. The color bar is non-linear by setting $\mathrm{sgn}(G_{\mathrm{R,C}})|G_{\mathrm{R,C}}|^{1/4}$.

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
