# Peer review of "Interference dynamics of matter-waves of SU($N$) fermions"

_SciPost Physics, doi:SciPost Phys. 15, 181 (2023)_

## Round 1 · Referee Report · Anonymous (Referee 1) · 2022-12-5

Strengths

1- The subject is timely 2- The study of SU(N) well complements corresponding studies for bosons and for 2-component fermioni c gases in rings 3- The study of momentum distribution and interferograms is well integrated 4- The article is overall clear

Weaknesses

1- The main weakness in my opinion is that the study of attractive interactions is only partially performed 2- The summary of literature on attractive SU(N) fermions is too compact, and it would be useful together with a more detailed study of the attractive interactions case

Report

The authors study the interference dynamics of SU(N) potentials in ring potentials. The paper is well motivated and clearly written, with the results well explained. My main remark is anyway that, given the clear motivation of the problem, the attractive interactions case should have been more expanded:

-) In the section II it is written

"For strong attractive interactions (U<<t) SU(N ) fermions are able to form bound states of different types and nature, which in turn causes part of the particles to localize together, while still adhering to the Pauli exclusion principle."

While this is true, I think a more detailed description of the phase diagram(s) of attractive SU(N) fermions is needed, see the point below. Notice that in the previous sentence the parenthesis U<<t should read |U|>>t if I am not wrong.

2) When in section III attractive interactions are considered, plots are discussed, but U is not varied. I think a discussion of different values of U would be important in section 3 [probably also in section 4, but that I can understand it could be defered to a subsequent publication], also in connection with the previous point 1).

3) Morever, for the two-component case the transition to the Tonks-Giraredeau gas has been discussed in literature, as in Fuchs, Recati and Zwerger, PRL (2004) and other papers, that helps to understand the quasi-1d case. A similar discussion for the SU(N) case with N>2 should be improved/provided, at least qualitatively.

If the previous remarks are taken into account, the readability and significance would improve and the paper could be published in my opinion.

Requested changes

1- Better discuss the results for SU(N) with attractive interactions in literature in the 1d and quasi-1d cases.

2- Improve the discussions of momentum distribution (and possibly of spiral interferograms) as a function of |U|.

---

## Round 1 · Referee Report · Kyrylo Snizhko (Referee 2) · 2022-12-15

Strengths

  1. Comprehensive study.

Weaknesses

  1. Mixing results and their explanations.
  2. Not enough information in the main text for a non-specialist to comfortably read the paper.

Report

In the present work, the authors investigate ultracold fermionic gases in toroidal traps. They investigate both non-interacting and interacting (both attractive and repulsive) gases. The authors derive predictions for interference experiments. Namely, for homodyne interference which gives access to the momentum distribution of the gas particles and heterodyne interference, which may produce spiral interference patterns, linked to the total angular momentum of the gas.

The main novelty of the consideration is that the authors study the gas in which fermions possess an SU(N) internal degree of freedom (as opposed to SU(2) studied previously). For homodyne detection, the authors focus on investigating the appearance of the dip in the center of the interferogram. For heterodyne detection, the authors focus on the appearance of glitches in the spiral. Both as a function of the flux penetrating the center of the system.

This is a comprehensive study, which may be useful for analyzing experiments in ultracold gas systems and verifying exotic properties of strongly-interacting systems. At the same time, the scope of the results corresponds rather to SciPost Physics Core (PRB-level) rather than SciPost (PRL-level) results.

Further, the paper in its present form is hard to understand unless one is very familiar with ultracold atoms and with strongly-interacting 1D systems. It is very hard to disentangle the results from attempts to explain them on the go. I strongly recommend the authors to work on making the paper more reader-friendly. I mention below a few important aspects of how this can be done.

1. In the abstract, the authors mention fractional quantization of the angular momentum. I understand that this is some sort of slang. The wave function of each particle should be single-valued as a function of the polar angle phi, leading to integer quantization of the total angular momentum – fractional quantization of the angular momentum seems unphysical from this point of view. On the other hand, if the authors prefer considering twisted boundary conditions due to the inserted flux, the angular momentum of such a “twisted” wave function can be quantized to fractional values even for non-interacting systems. If the statement is crucial to the authors, it should be explained early on in the paper. On the contrary, if the statement is not essential to the main message of the paper, it should not appear in the abstract – where it raises doubts about the validity of the work.

2. Early in section III.A, the authors refer to n, quantum numbers of the levels occupied by particles. While this is a free particle system and one can guess that n corresponds to momentum quantization along the chain, this is not spelled out explicitly. In connection to total angular momentum l appearing in the same paragraph, this makes the reading somewhat confusing. Further, the authors refer to parabolas of different angular momentum – it would be most reader-friendly to provide a figure of such parabolas and how changing the flux makes the system move from one parabola to another. The authors actually provide the figures and the explanations in Appendix A.1. I find that having them in the main text would make the paper significantly easier to read.

3. In the last paragraph of III.A, the authors write about W, the ratio of the number of particles in the system to the number of fermionic flavours. In particular, the authors discuss the difference between “equal and commensurate value” of W and “the case when W is commensurate but not equal for different SU(N)”. I can guess that what is meant here is the number of fermions occupying each flavour being an integer multiple of the chain length, yet possibly not the same for different flavours. However, providing a slightly more extended discussion, where the reader would be given a proper definition, would be beneficial. It would also be nice to briefly discuss why equilibration within each flavour is assumed, but no equilibration between flavours is considered possible.

4. In section III.B.1, when discussing repulsive interactions, it would be beneficial to discuss the origin and the “fractionalized” parabolas in the main text, and not refer the reader to the Appendix.

An alternative approach to presenting the paper results could be highlighting the key novel results without explaining their origin. And then derive the results/provide their microscopic explanations later in the paper/in the Appendix. The present way of presenting, when the result explanation is expected to be understandable, but there is not enough information to actually understand it, makes the text very uncomfortable to read.

A few smaller remarks.

In the introduction, the authors write: “ultracold atoms feature robust coherence properties withour cryogenics”. According to some definitions (e.g., https://en.wikipedia.org/wiki/Cryogenics), cryogenics is defined by studying low temperatures – not necessarily by using cooling liquids to achieve those. The authors may want to reformulate the phrase.

In the paragraph after the one containing Eq. (1), the authors write: “for strong attractive interactions (U << t)”. I presume U < 0, |U| >> t is meant.

In Eqs. (A14-A17), a discrete sum formula is connected to the Bessel functions represented through a continuous integral (A15, by the way, the differential is missing from the integral). Is the connection valid for any chain length L or only in the thermodynamic limit L->infty?

Requested changes

  1. Improve the presentation of the paper following one of the routes suggested in the report.

---

## Round 2 · Referee Report · Kyrylo Snizhko (Referee 2) · 2023-2-24

Strengths

Comprehensive consideration of various interaction strengths

Weaknesses

Insufficient attention to defining the relevant terms and physical quantities necessary for understanding the underlying physics.

Not clear to what extent the conclusions for small particle numbers in interacting systems are extendable to large particle numbers present in experiments.

Report

I thank the authors for their corrections and clarifications. The manuscript has clearly been improved. At the same time, I still do not find it ready for being published.

The key issue for me is still “fractional angular momentum”. In Sec. 3.1, the authors define $\ell$ as “the total angular momentum”. In appendix A.2, they define the same $\ell$ as “the angular momentum per particle”. From the behavior outlined in Sec. 3.1, it appears to me that in both contexts $\ell$ is the angular momentum per particle.

Then the ability of $\ell$ to acquire fractional values is not surprising. Consider a non-interacting system with a single smallest particle-hole excitation: the total angular momentum is increased by 1, while the angular momentum per particle is increased by $1/N_p$. Therefore, in the presence of interactions, when the ground state does not correspond to a uniformly filled Fermi sphere, it is only natural to expect fractional values of the angular momentum per particle. The miracle would be if these fractional values were quantized instead of forming a smooth continuum. Based on the statement in the beginning of Sec. 3.2.1 and the energy-flux diagram in Fig. 12, for infinitely-repulsive interactions in the thermodynamic limit the “quantization step” of the angular momentum per particle tends to zero; i.e., the fractional values do form a smooth continuum.

Given this, I strongly oppose the phrasing “orbital angular momentum … quantized to fractional values” appearing in the abstract. Stating a reduced period of persistent current’s dependence on flux would be a much clearer way of describing the statement that makes the system interesting to the authors.

A technical remark on this issue: the newly-introduced relation between $\ell$, $n$, and $I_j$ in the beginning of Appendix A.2 does not help – and rather harms: an unsuspecting reader suddenly learns of existence of $I_j$ which have never been defined before. I would recommend removing this piece of text: the total angular momentum per particle is a much more understandable observable than Bethe-ansatz quantum numbers. Further, there is no need to introduce Bethe-ansatz quantum numbers when talking in terms of non-interacting particles (“Fermi sphere displaced by”).

Concerning the relevance of the paper to the broad audience. The paper is, as I wrote before, a comprehensive study of the signatures produced for various interaction strengths. However, the authors perform studies for interacting systems with the number of particles $\leq 10$. This is understandable from the point of view of the theory – interacting systems are hard to study. However, it is not clear to what extent the conclusions extend to larger numbers of particles. At the same time, the experiments of Refs. 4 and 5 from the author’s response letter (Refs. 41 and 42 in the manuscript) use $N_p$ of the order of 10 000.

Given this uncertainty of whether the predictions are relevant for the large numbers of particles in experiments, and given that the paper is rather hard to read to an advanced non-specialist (that is, myself), I would recommend publishing the paper in a more specialized journal. For specialists, it would be an important advance and a basis for further development, while the jargon concerning “fractionalization” and assumed understanding of the definitions would constitute less of a problem.

Given the acceptance criteria of SciPost Physics (https://scipost.org/SciPostPhys/about#criteria) and SciPost Physics Core (https://scipost.org/SciPostPhysCore/about#criteria), I would recommend publishing the paper in SciPostPhysics Core after the authors make the minor changes I request in the field "Requested changes".

A minor technical issue: reference to a figure (probably, Fig. 14) on page 22 is broken. (See the paragraph starting with “Attraction: For infinitely attractive interactions, the energy”).

Requested changes

Explain the issue of "fractional angular momentum"/"reduced period of the persistent current dependence on the flux" in a correct, self-consistent manner. Possibly, exclude unnecessary aspects.

Clarify in the manuscript to what degree the formula stated for general $N_p$ and $N$ are valid. State what this validity is based on.

---

## Round 2 · Referee Report · Anonymous (Referee 1) · 2023-3-8

Strengths

Timely subject

Well complementing similar studies for bosons

Interesting results for momentum distribution and interferograms

Improved clarity

Weaknesses

The study of attractive interactions is not complete

Report

I read the revised version. I continue to think that the study of attractive interactions is not yet complete, as also the authors comment about, but at the same time I think the paper significantly improved both in clarity and in addressing the points I mentioned. The topic is interesting and timely, and the discussion of the fingerprints of the different regimes in the considered setup with N components rather readable. For these reasons I am in favour of the publication.

Requested changes

No further requested changes.

---

## Round 2 · Author Response

Reply to the Referee reports is included alongside the re-submitted manuscript, where the changes are highlighted in red.

---

## Round 3 · Referee Report · Kyrylo Snizhko (Referee 2) · 2023-5-12

Strengths

Interesting results that raise interesting questions

Weaknesses

Multiple results for different observables are piled up

Results are poorly linked together: unclear, how the reduced period in flux dependence, e.g., in the persistent current, is physically related to a delay in flux dependence of another observable

Results are not clearly sorted according to whether they are relevant for experiments with small/large particle numbers

Report

Warnings issued while processing user-supplied markup:

  • Inconsistency: Markdown and reStructuredText syntaxes are mixed. Markdown will be used.
    Add "#coerce:reST" or "#coerce:plain" as the first line of your text to force reStructuredText or no markup.
    You may also contact the helpdesk if the formatting is incorrect and you are unable to edit your text.

I thank the authors for responding to my comments and for making corrections to the manuscript. The authors have brought it to a point, where I feel that the manuscript may be published. But not in SciPost Physics, rather in SciPost Physics Core.

Below, I explain the grounds for such a recommendation. I start with responding to some of the authors’ comments in their response letter.

The authors state: “we note that the persistent current, being a genuine mesoscopic quantity, is identically zero in the thermodynamic limit and therefore the ‘smooth continuum’ cannot be observed in that limit”. This is evidently wrong. The most obvious counterexample of a persistent current on macroscopic scales is given by superconducting levitation experiments: https://www.youtube.com/watch?v=AWojYBhvfjM. Even if one restricts the attention to cold atomic systems, Ref. 39 performed experiments with about 60 000 atoms. In which case $1/N_p < 10^{-4}$, leading to the possibility of a quasicontinuum to quite a good approximation.

The authors state: “If we understand correctly, the Referee makes an argument on the ability of the angular momentum per particle to acquire fractional values based on particle-hole excitations in free and weakly interacting systems. Their argument would imply that any weakly interacting system would display 1/Np angular momentum fractionalization. This is in clear contrast with the experimental findings (Ref. 39 in the manuscript) in which the angular momentum per particle in a quantum fluid is indeed quantized irrespective of the number of particles.”

My argument indeed states that having a fractional values of angular momentum per particle is not a wonder. Moreover, the lower limit on the fractional values that can occur in a system with $N_p$ particles is given by $1/N_p$. Why does then Ref. 39 not observe such fractional values? Note that the system is cooled to its ground state or close to it. So, the observed deviation from an integer value should be proportional to $N_{ex}/N_p$, where $N_{ex}$ is the number of particles excited above the non-interacting ground state (due to either imperfect cooling or the change in the ground state due to the weak interactions). If the number of excited particles is < 10%, the deviation lies well within the error bars in Fig. 3. Further, for large $\Omega$, when the cooling may break down, the angular momentum per particle varies continuously with $\Omega$ in qualitative agreement with my argument.

In other words, assuming perfect cooling to the ground state, I would expect integer quantization at vanishing interactions, and near-integer values for weak interactions. Which is what was observed in Ref. 39.

Appearance of robust fractional steps would indeed be surprising. However, the authors of the present manuscript do not make this claim. Angular momentum per particle in the interacting system is never calculated in the manuscript. The link between the calculated experimentally observable quantities and the angular momentum per particle is only implied by the text, and the implication is based on the intuition coming from non-interacting systems.

What the authors consider, is the dependence of some observable quantities on the flux threading the system. Which is why I repeat my opinion from the previous report: I strongly oppose the phrasing “orbital angular momentum ... quantized to fractional values” appearing in the abstract. Saying that interactions change the flux dependence of the observables in an unexpected way, would be a much clearer and a much more solid statement.

The paper contains a number of results concerning the flux dependence of observables. The citation of Ref. 43 and Fig. 12 appearing in the paper’s appendix state that the flux periodicity of some observables should be inversely propotional to the number of particles, when the interactions are infinitely strong. This is what led to the above discussion concerning the thermodynamic limit. I acknowledge the author’s remark that systems with small number of particles, where the effect of steps should be visible, can, in principle, be studied.

Figure 3 in the main text says something different about a specific observable. Namely, the appearance of a dip in the density distribution is delayed by $1/2-1/(2N_p)$ for strong interactions. $1/2-1/(2N_p)$ tends to 1/2 in the thermodynamic limit and is thus observable for a large number of particles. In my opinion, this can be called fractionalization (not clear, though, of what). Unfortunately, this formula is only hinted at by the results for a small number of particles. Moreover, the physics behind this effect (the relation to the reduced periodicity, the relation to angular momentum per particle etc.) is not clear.

For the attractive interactions, the effects scale with the number of species, which makes them suitable to be observed in the thermodynamic limit. Figure 5 shows a reduced flux interval between jumps. And Fig. 4 shows a similar reduction of the flux intervals between features. Again, these are results for a small number of particles. Their generalizability and physical origin are not clear.

One sees, that the paper does contain a number of interesting results. However, there are two problems: - The language of angular momentum fractionalization may be highly misleading (may not be too – but the paper does not present enough evidence and explanations to make the judgement); - The paper does not present physical insight into the origin of the discussed effects and thus will hardly be understandable to a broader audience. When properly interpreted, the results might indeed spark multipronged follow-up work in the professional subcommunity (three directions being theoretical clarification of what’s going on, and constructing experiments for small and large number of particles); but this proper interpretation is extremely hard to read in the paper.

Therefore, I have two recommendations. I suggest the authors to avoid the language of “angular momentum fractionalization” in order not to mislead the reader. I suggest the editor to accept the paper to SciPost Physics Core, where it can serve as a reference to the professional subcommunity.

  • validity: high
  • significance: good
  • originality: high
  • clarity: ok
  • formatting: perfect
  • grammar: perfect

Author:  Wayne Jordan Chetcuti  on 2023-05-18  [id 3680]

(in reply to Report 1 by Kyrylo Snizhko on 2023-05-12)

R:“I thank the authors for responding to my comments and for making corrections to the manuscript. The authors have brought it to a point, where I feel that the manuscript may be published. But not in SciPost Physics, rather in SciPost Physics Core.

Below, I explain the grounds for such a recommendation. I start with responding to some of the authors’ comments in their response letter. ”

  1. R: The authors state: “we note that the persistent current, being a genuine mesoscopic quantity, is identically zero in the thermodynamic limit and therefore the ‘smooth continuum’ cannot be observed in that limit”. This is evidently wrong. The most obvious counterexample of a persistent current on macroscopic scales is given by superconducting levitation experiments: https://www.youtube.com/watch?v=AWojYBhvfjM. Even if one restricts the attention to cold atomic systems, Ref. 39 performed experiments with about 60 000 atoms. In which case, 1/Np < 10−4, leading to the possibility of a quasicontinuum to quite a good approximation.”

A: The confusion in the mind of the Referee arises since an erroneous understanding of what thermodynamic limit means: Number of particles & system size going to infinity in such a way that the ratio between the two quantities is finite.

Persistent currents result whenever the ring’s spatial scale is comparable to the particles’ coherence length. As it is well known in mesoscopic physics, the persistent current scales as L1 (see Equation 4.6 page 69 in the book “Introduction to mesoscopic physics” by Joe Imry). For the particular case of cold atoms mentioned by the Referee, persistent currents have been observed experimentally with systems having around 105 atoms in rings that have a size to the order of μm. Such system is mesoscopic. We remark that the fact that the persistent current vanishes in the thermodynamic limit can also be seen in the reference mentioned in the manuscript (see, for instance, the expression of the current Equation 2 in Ref 43.) where one can see that there is an inverse relationship with the length of the system.

We point out that the experiment in Ref. 39 was carried out for bosons and not fermions —- persistent currents of repulsive bosons have a corresponding angular momentum quantized to integer values (see Figure 3 in Ref. 39.).

  1. R: “The authors state: “If we understand correctly, the Referee makes an argument on the ability of the angular momentum per particle to acquire fractional values based on particle-hole excitations in free and weakly interacting systems. Their argument would imply that any weakly interacting system would display 1/Np angular momentum fractionalization. This is in clear contrast with the experimental findings (Ref. 39 in the manuscript) in which the angular momentum per particle in a quantum fluid is indeed quantized irrespective of the number of particles.”

My argument indeed states that having a fractional values of angular momentum per particle is not a wonder. Moreover, the lower limit on the fractional values that can occur in a system with Np particles is given by 1/Np. Why does then Ref. 39 not observe such fractional values? Note that the system is cooled to its ground state or close to it. So, the observed deviation from an integer value should be proportional to Nex/Np, where Nex is the number of particles excited above the non-interacting ground state (due to either imperfect cooling or the change in the ground state due to the weak interactions). If the number of excited particles is ¡ 10%, the deviation lies well within the error bars in Fig. 3. Further, for large Ω, when the cooling may break down, the angular momentum per particle varies continuously with Ω in qualitative agreement with my argument.

In other words, assuming perfect cooling to the ground state, I would expect integer quantization at vanishing interactions, and near-integer values for weak interactions. Which is what was observed in Ref. 39.

Appearance of robust fractional steps would indeed be surprising. However, the authors of the present manuscript do not make this claim. Angular momentum per particle in the interacting system is never calculated in the manuscript. The link between the calculated experimentally observable quantities and the angular momentum per particle is only implied by the text, and the implication is based on the intuition coming from non-interacting systems.

What the authors consider, is the dependence of some observable quantities on the flux threading the system. Which is why I repeat my opinion from the previous report: I strongly oppose the phrasing “orbital angular momentum ... quantized to fractional values” appearing in the abstract. Saying that interactions change the flux dependence of the observables in an unexpected way, would be a much clearer and a much more solid statement.”

A: The Referee’s whole argument hinges on the fact that in the experiment of Ref. 39, the persistent current does not display fractional steps reflecting the fractional quantization of the angular momentum. The problem of the argument may arise from the fact that he seems to apply a classical physics reasoning to a quantum fluid. On top of this, we reiterate that we are considering fermions and not bosons.

A very well known fact of quantum many-body systems is that the response of the ground-state to the rotation is called the angular momentum. For example, the Referee can check the Dalibard lectures in Varenna (see Equation 24 in http://www.phys.ens.fr/~dalibard/publications/2015_Varenna_JD.pdf ).

  1. R: “The paper contains a number of results concerning the flux dependence of observables. The citation of Ref. 43 and Fig. 12 appearing in the paper’s appendix state that the flux periodicity of some observables should be inversely propotional to the number of particles, when the interactions are infinitely strong. This is what led to the above discussion concerning the thermodynamic limit. I acknowledge the author’s remark that systems with small number of particles, where the effect of steps should be visible, can, in principle, be studied.

Figure 3 in the main text says something different about a specific observable. Namely, the appearance of a dip in the density distribution is delayed by 1/2 − 1/(2Np) for strong interactions. 1/2 − 1/(2Np) tends to 1/2 in the thermodynamic limit and is thus observable for a large number of particles. In my opinion, this can be called fractionalization (not clear, though, of what). Unfortunately, this formula is only hinted at by the results for a small number of particles. Moreover, the physics behind this effect (the relation to the reduced periodicity, the relation to angular momentum per particle etc.) is not clear.

For the attractive interactions, the effects scale with the number of species, which makes them suitable to be observed in the thermodynamic limit. Figure 5 shows a reduced flux interval between jumps. And Fig. 4 shows a similar reduction of the flux intervals between features. Again, these are results for a small number of particles. Their generalizability and physical origin are not clear.”

A: The above argument is based on the wrong application of a formula that we provided. We specifically indicate that the delay for the appearances is given by φ_{H} + (N_{p}−1)/(2N_{p}), where φ_{H} is the flux at which a hole appears for the system with zero interactions and Np is the number of particles.

  1. R:“One sees, that the paper does contain a number of interesting results. However, there are two problems:

• The language of angular momentum fractionalization may be highly misleading (may not be too – but the paper does not present enough evidence and explanations to make the judgement);

• The paper does not present physical insight into the origin of the discussed effects and thus will hardly be understandable to a broader audience. When properly interpreted, the results might indeed spark multi- pronged follow-up work in the professional subcommunity (three directions being theoretical clarification of what’s going on, and constructing experiments for small and large number of particles); but this proper interpretation is extremely hard to read in the paper.”

A: We strongly disagree with the Referee. We produce arguments and references that we apply the correct language in the manuscript. On top of that, our results are perfectly understandable with basic notions in mesoscopic physics.

  1. R:“Therefore, I have two recommendations. I suggest the authors to avoid the language of “angular momentum fractionalization” in order not to mislead the reader. I suggest the editor to accept the paper to SciPost Physics Core, where it can serve as a reference to the professional subcommunity.”

A: In our opinion the paper is very suitable for SciPost Physics.

Kyrylo Snizhko  on 2023-05-30  [id 3696]

(in reply to Wayne Jordan Chetcuti on 2023-05-18 [id 3680])
Category:
reply to objection

The authors fail to trace the context of the discussion.

The first comment, concerning "$1/N_p < 10^{−4}$" concerned the issue that the angular momentum per particle can very quasicontinuously. Even if $Np = 105$, the step $1/N_p < 10^{-2}$ is pretty quasicontinuous, in my view. (Probably, the authors meant $N_p=10^{5}$, which makes my argument even stronger and the step $1/N_p$ is then even smaller).

The second comment, concerning Ref. 39 was in response to the authors' statement that Ref. 39 did not observe fractional values. I have provided an explanation that it has - but not quantization to fractional values. The author's response does not address this issue except for the statement that they consider fermions as opposed to bosons in Ref. 39. However, it is the authors' result for fermions that the angular momentum per particle is "quantized" in $1/N_p$, unless the interactions are attractive.

Given that the authors fail to trace the context of the discussion, I see no value in continuing it. I have said all I can. The final decision is to be made by the editor.

---

## Round 3 · Referee Report · Anonymous (Referee 3) · 2023-9-27

Strengths

1- All the results are sound and clearly discussed in detail.

2- The underlying many-body theory is clearly formulated to address also a broader audience.

3- Well-written appendices are included to explain the different homodyne and heterodyne detection protocols.

4- The manuscript has been further improved during a relatively long submission process, in order to address the referee comments and criticism.

Weaknesses

None

Report

In this work the authors characterize the fractionalization of the persistent current flowing in a SU(N) fermionic circuit in terms of time-of-flight interference patterns, when varying from free particles to the case of repulsive and attracting interactions. To do so, they successfully combine analytical, as exact diagonalization, and numerical techniques, as DMRG. The very interesting achieved results can be experimentally tested with the state-of-the-art cold atom infrastructures. Given also the strengths outlined above, I recommend the publication of this manuscript in SciPost Physics.

Requested changes

No requested changes.

---

## Round 3 · Author Response

Reply to the Referee reports is included alongside the re-submitted manuscript, where the changes are highlighted in blue.

---

## Editorial Decision

published